**METHOD**                                                                                    **Open Access**

# Lisa: inferring transcriptional regulators through integrative modeling of public chromatin accessibility and ChIP-seq data

Qian Qin[1,2†], Jingyu Fan[1†], Rongbin Zheng[1], Changxin Wan[1], Shenglin Mei[1], Qiu Wu[1], Hanfei Sun[1], Myles Brown[5,6], Jing Zhang[3*], Clifford A. Meyer[4,6] ![ORCID] and X. Shirley Liu[4,6*]

## Abstract

We developed Lisa (http://lisa.cistrome.org/) to predict the transcriptional regulators (TRs) of differentially expressed or co-expressed gene sets. Based on the input gene sets, Lisa first uses histone mark ChIP-seq and chromatin accessibility profiles to construct a chromatin model related to the regulation of these genes. Using TR ChIP-seq peaks or imputed TR binding sites, Lisa probes the chromatin models using in silico deletion to find the most relevant TRs. Applied to gene sets derived from targeted TF perturbation experiments, Lisa boosted the performance of imputed TR cistromes and outperformed alternative methods in identifying the perturbed TRs.

**Keywords:** Transcription factors, Gene regulation, Chromatin accessibility, DNase-seq, H3K27ac ChIP-seq, Differential gene expression, Gene set analysis

## Introduction

Transcriptional regulators (TRs), which include transcription factors (TFs) and chromatin regulators (CRs), play essential roles in controlling normal biological processes and are frequently implicated in disease [1–4]. The genomic landscape of TF binding sites and histone modifications collectively shape the transcriptional regulatory environments of genes [5–8]. ChIP-seq has been widely used to map the genome-wide set of *cis*-elements bound by *trans*-acting factors such as TFs and CRs, which we henceforth refer to as "cistromes" [9]. There are approximately 1500 transcription factors in humans and mice [10, 11], regulating a wide variety of biological processes in constitutive or cell-type-specific manners, and tens of thousands of ChIP-seq and DNase-seq experiments have been performed in humans and mice. We previously developed the Cistrome Data Browser

(DB) [12], a collection of uniformly processed TF ChIP-seq (~ 11,000) and chromatin profiles (~ 12,000 histone mark ChIP-seq and DNase-seq) in humans and mice.

The question we address in this paper is how to effectively use these data to infer the TRs that regulate a query gene set derived from differential or correlated gene expression analyses in humans or mice. TR ChIP-seq data, when available, is the most accurate available data type representing TR binding. ChIP-seq data availability, in terms of covered TRs and cell types, even with large contributions from projects such as ENCODE [13], is still sparse due to the limited availability of specific antibodies. Although advances have been made in TR cistrome mapping with the introduction of technologies such as CETCh-seq [14] and CUT & RUN [15], the difficulties in acquiring TR ChIP-seq data for new factors limit the TR by cell type coverage of high-quality TR ChIP-seq data. Chromatin accessibility data, including DNase-seq [16, 17] and ATAC-seq [18], is available for hundreds of cell types and provides maps of the regions in which TRs are likely to be bound in the represented cell types. The H3K27ac histone modification, associated with active enhancers and promoters of actively transcribed genes, has been widely profiled using ChIP-seq in many cell types [5, 19]. When TF ChIP-seq data is

* Correspondence: zhangjing@tongji.edu.cn; cliff_meyer@ds.dfci.harvard.edu; xsliu@ds.dfci.harvard.edu
†Qian Qin and Jingyu Fan contributed equally to this work and can interchangeably be ordered as co-first authors.
³Stem Cell Translational Research Center, Tongji Hospital, School of Life Science and Technology, Tongji University, Shanghai 200065, China
⁴Center for Functional Cancer Epigenetics, Dana-Farber Cancer Institute, Boston, MA 02215, USA
Full list of author information is available at the end of the article

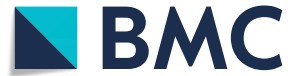

not available, TF binding motifs, used in combination with chromatin accessibility data or H3K27ac ChIP-seq data might be used to infer TF binding sites [7, 20, 21]. Machine learning approaches that transfer models learned from TF ChIP-seq peaks, motifs, and DNase-seq data between cell types are promising ways of imputing TF cistromes, although imputation of TF binding sites on a large scale remains to be implemented [22–27]. Computationally imputed TF binding data is expected to represent TF binding sites less accurately than TF ChIP-seq experimental data, so we sought to develop a TR prediction method that could use imputed TF cistromes effectively, along with ChIP-seq-derived ones.

We previously developed MARGE to characterize the regulatory association between H3K27ac ChIP-seq and differential gene expression in terms of a regulatory potential (RP) model [28]. The RP model provides a summary statistic of the *cis*-regulatory influence of the many *cis*-regulatory elements that might influence a gene's transcription rate. MARGE builds a classifier based on H3K27ac ChIP-seq RPs from the Cistrome DB to discriminate the genes in a query differentially expressed gene set from a set of background genes. One of the functions of MARGE is to predict the *cis*-regulatory elements (i.e., genomic intervals) that regulate a gene set. BART [29] extends MARGE, to predict the TRs that regulate the query gene set through an analysis of the predicted *cis*-regulatory elements. Here, we describe Lisa (epigenetic Landscape In Silico deletion Analysis and the second descendent of MARGE), a more accurate method of integrating H3K27ac ChIP-seq and DNase-seq with TR ChIP-seq or imputed TR binding sites to predict the TRs that regulate a query gene set. Unlike BART, Lisa does not carry out an enrichment analysis of the *cis*-regulatory elements predicted by MARGE. Instead, Lisa analyses the relationship between TR binding and the gene set using RP models and RP model perturbations. We assessed the performance of Lisa and other TR identification methods, BART [29], i-cisTarget [30], and Enrichr [31], using differentially expressed gene sets derived from experiments in which the activities of specific TFs were perturbed by knockdown, knockout, over-expression, stimulation, or inhibition.

## Results and discussion
### Regulatory TR prediction based on Cistrome DB ChIP-seq peaks
High-quality TR ChIP-seq data, when available, accurately characterizes genome-wide TR binding sites, which can be used to infer the regulated genes in particular cell types. Estimating the effect of TR binding on gene expression is not trivial because: (1) there is no accurate map linking enhancers to the genes they regulate [32]; (2) multiple enhancers can regulate the same gene [33],

and a single enhancer can regulate multiple genes [34]; and (3) not all TR binding sites are functional enhancers [19]. A model is therefore needed to quantify the likelihood of a gene being regulated by a TR cistrome. The "peak-RP" model [35, 36] is based on TR ChIP-seq peaks, serving as a proxy for TR binding sites, without the use of DNase-seq or H3K27ac ChIP-seq data. In the peak-RP model (Fig. 1a), the effect a TR binding site has on the expression of a gene is assumed to decay exponentially with the genomic distance between the TR binding site and the transcription start site (TSS), and the contribution of multiple binding sites is assumed to be additive [36]. Accounting for the number of TR binding sites and for the distances of these sites from the TSS has been shown to be more accurate than alternative TR target assignment methods [37]. While it is possible that enhancers could modulate each other in non-additive ways [32], data on these types of behavior are too scarce to incorporate in a TR prediction model.

We use the peak-RP model to identify TFs that are likely regulators of a target gene set by searching for Cistrome DB [12] cistromes that produce higher peak-RPs for the query gene set than for a set of background genes (Additional file 1: Figure S1, Additional file 2: Table S1). Statistical significance is calculated using the one-sided Wilcoxon rank-sum test statistic comparing the peak-RPs for the query gene set with the background. The TRs with the most significant $p$ values are considered to be the candidate regulators. Lisa uses TR ChIP-seq within the peak-RP model, along with the chromatin landscape models described below to infer the TRs of a gene set.

### Regulatory TR prediction using a chromatin landscape model
While TR ChIP-seq data provides accurate information about TR cistromes in specific cell types, the Cistrome DB TR by cell type coverage is skewed towards a few TRs, such as CTCF, which are represented in many cell types, and towards cell types such as K562 (Additional file 1: Figure S1b-c), in which many TRs have been characterized (Additional file 1: Figure S1d). H3K27ac ChIP-seq [19] and DNase-seq [16], available in a large number and variety of cell types, can be used to infer cell-type-specific regulatory regions. These types of data could enhance the use of TR ChIP-seq data as well as imputed TF binding data, which may not accurately represent TF binding sites in different cell contexts.

To boost the performance of TF ChIP-seq or imputed TF binding data in the identification of regulatory TRs, we developed Lisa chromatin landscape models, which use H3K27ac ChIP-seq and DNase-seq chromatin profiles (Fig. 1b, Additional file 3: Table S2; see the "Methods" section) to model the regulatory importance

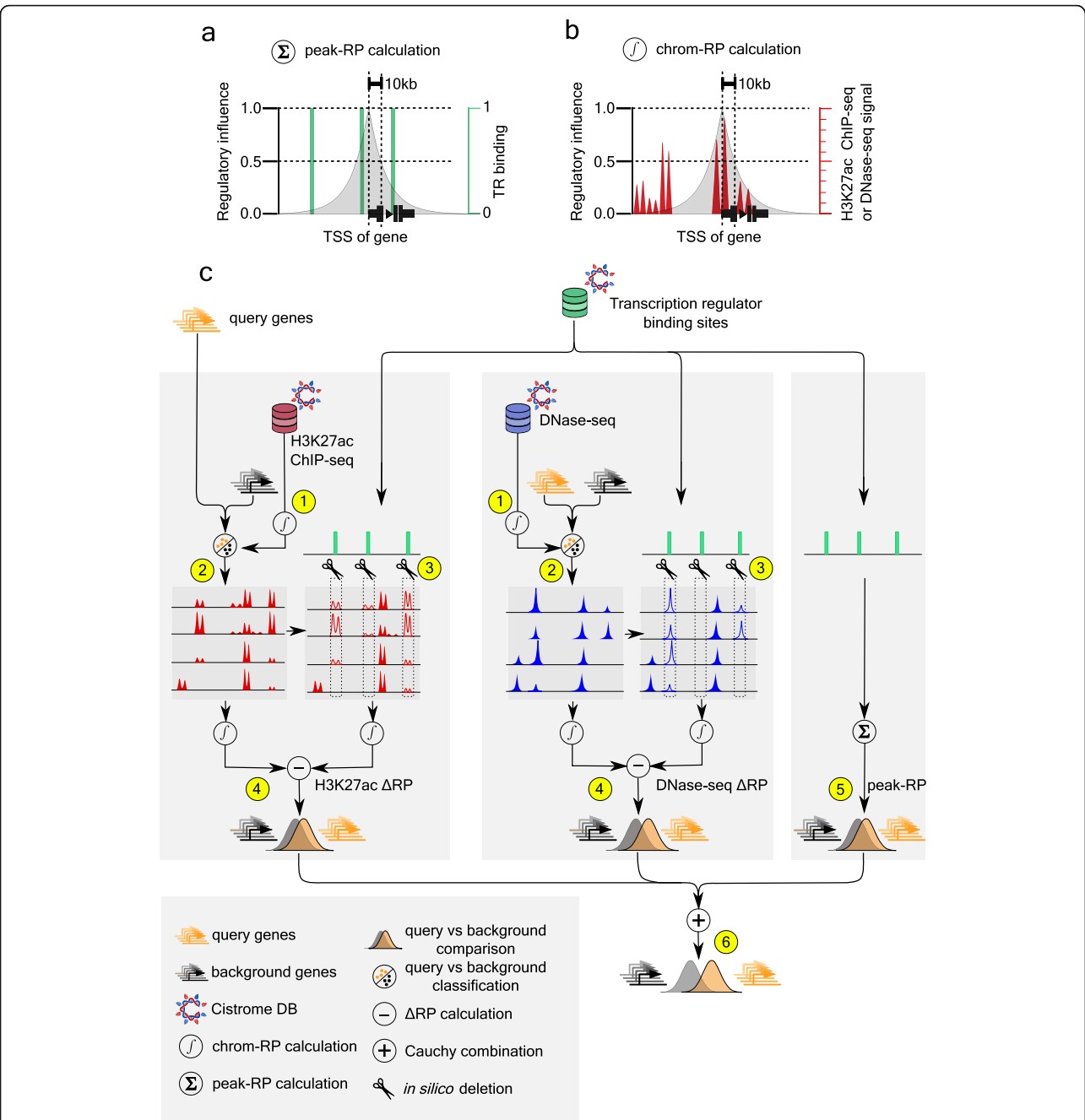

**Fig. 1** Illustration of the Lisa framework. **a** The peak-RP score models the effect of TR binding sites on the regulation of a gene. TR binding sites are binary values, and peaks nearer to the gene's TSS have a greater influence than ones further away. **b** The chrom-RP score summarizes the effect of the DNase-seq or H3K27ac chromatin environment on a gene. The chrom-RP score is based on a continuous rather than a binary signal quantification. **c** Overview of the Lisa framework. (1) H3K27ac ChIP-seq or DNase-seq data from the Cistrome DB is summarized using the chrom-RP score for each gene. (2) H3K27ac ChIP-seq or DNase-seq samples that can discriminate between the query gene set and the background gene set are selected, and the regression parameters define a chrom-RP model. (3) Each TR cistrome from the Cistrome DB is evaluated as a putative regulator of the query gene set through in silico deletion, which involves the elimination of H3K27ac ChIP-seq or DNase-seq signal at the binding sites of the putative regulator. (4) The chrom-RP model, based on in silico deletion signal, is compared to the model without deletion for each gene in the query and background gene sets. A *p* value is calculated using the Wilcoxon rank test comparison of the query and background ΔRPs. (5) The peak-RP based on TR ChIP-seq peaks is calculated for the putative regulatory cistrome, and the statistical significance of peak-RP distributions from the query and background gene sets is calculated. (6) *p* values from the H3K27ac ChIP-seq, DNase-seq, and peak-RP analysis are combined using the Cauchy combination test. TR cistromes are ranked based on the combined *p* value

of different genomic loci. As differential gene expression experiments are not always carried out in parallel with chromatin profiling experiments, Lisa does not require the corresponding user-generated chromatin profiles but instead uses the DNase-seq and H3K27ac ChIP-seq data that is available in the Cistrome DB to help identify *cis*-regulatory elements controlling a differential expression gene set. To this end, Lisa models chromatin landscapes through chromatin RPs (chrom-RPs, Fig. 1b), which are defined in a similar way to the peak-RP with one small difference: genome-wide read signals instead of peak calls are used in the calculation of the chrom-RP [28]. Changes in H3K27ac ChIP-seq and DNase-seq associated with cell state perturbations are often a matter of degree rather than switch-like; therefore, we base the chrom-RP on reads rather than peaks. The chrom-RP is pre-calculated for each gene (Fig. 1c (1)) and for each H3K27ac ChIP-seq/DNase-seq profile in the Cistrome DB (Additional file 1: Figure S1a, Additional file 3: Table S2). These chrom-RPs quantify the *cis*-regulatory activities that influence each gene under cell-type-specific conditions.

Given the query gene set, Lisa identifies a small number of Cistrome DB DNase-seq and H3K27ac ChIP-seq samples that are informative about the regulation of these genes. Lisa does this by using the pre-calculated H3K27ac/DNase-seq chrom-RPs to discriminate between the query gene set and a background gene set. Using L1-regularized logistic regression, Lisa assigns a weight to each selected sample so that the weighted sum of the chrom-RPs on the genes best separates the query and the background gene sets (Fig. 1c (2)). This step is carried out separately for H3K27ac ChIP-seq and DNase-seq, yielding a chrom-RP model based on H3K27ac ChIP-seq and another model based on DNase-seq.

Next, by a process of in silico deletion (ISD), Lisa evaluates the effect deleting each TR cistrome has on the chromatin landscape model (Fig. 1c (3)). ISD of a TR cistrome involves setting DNase-seq or H3K27ac ChIP-seq chromatin signal to 0 in the 1-kb intervals containing the peaks in that cistrome and evaluating the effect on the predictions made by the chromatin landscape models. The difference of the model scores before ISD and after ISD quantifies the impact that the deleted TR cistrome is predicted to have on the query and background gene sets. Lisa does not make a prediction of *cis*-regulatory elements, the approach taken by MARGE and BART. Instead, Lisa probes the effects of deleting putative regulatory TR cistromes on the chrom-RP model. Whereas the chrom-RP integrates data over 200-kb intervals, the scale of individual *cis*-regulatory elements is of the order of 1 kb. The ISD approach mitigates the difficulties in transferring information contained in the chrom-RP model from the chrom-RP (200 kb) scale to the *cis*-regulatory element (1 kb) scale.

Finally, to prioritize the candidate TRs, Lisa compares the predicted effects on the query and background gene sets using the one-sided Wilcoxon rank-sum test (Fig. 1c (4)). A one-sided test is used because Lisa assumes that the in silico deletion of a true regulatory factor will decrease, not increase, the model's ability to discriminate between query and background gene sets. To utilize the power of predictions based on H3K27ac-ChIP-seq and DNase-seq ISD models, and TF ChIP-seq peak-only models (Fig. 1c (5)), the results are combined using the Cauchy combination test [38] (Fig. 1c (6)). Whereas MARGE [28] predicts *cis*-regulatory elements (but does not analyze TRs), and BART [29] carries out an enrichment analysis of predicted *cis*-elements to discover TRs, Lisa uses the chromatin landscape model in a different way. In combination with ChIP-seq-derived or computationally imputed TR binding, Lisa probes the effects of TRs on the chromatin RP models of query and background gene sets.

## Demonstration of chromatin landscape models in a GATA6 knockdown study

We demonstrate Lisa chromatin landscapes and in silico deletion using a query gene set defined as the downregulated genes in a GATA6 knockdown experiment in the KATO-III stomach cancer cell line [39] (Fig. 2). Lisa identifies DNase-seq and H3K27ac ChIP-seq chromatin landscape models (Fig. 2a, Fig. 1c (2)), which include several gastro-intestinal samples (Additional file 1: Figure S2b,d) whose chrom-RPs can discriminate between the query and background gene sets (Additional file 1: Figure S2a, DNase-seq ROC AUC = 0.816, Additional file 1: Figure S2c, H3K27ac ROC AUC = 0.821). In silico deletion (Fig. 1c (3)) of GATA6 binding sites produces larger DNase-seq and H3K27ac $\Delta$RPs (DNase $\Delta$RP, 1.05; H3K27ac $\Delta$RPs, 0.25) for an example downregulated gene, *LINC01133* [40], than for a background gene, *ZC3H12A* (DNase $\Delta$RP, 0.06; H3K27ac $\Delta$RP, 0.01) (Fig. 2b). In silico deletion of CTCF binding sites, in contrast, has a smaller effect on the chromatin landscapes surrounding *LINC01133* (DNase $\Delta$RP, 0.02; H3K27ac $\Delta$RP, 0.01), resulting in $\Delta$RPs that are more similar to the $\Delta$RPs for *ZC3H12A* (Fig. 2b) (DNase $\Delta$RP, 0.004; H3K27ac $\Delta$RP, 0.001). Statistical analysis is carried out comparing all the query gene $\Delta$RPs with all the background gene $\Delta$RPs (Fig. 1c (4)), producing significant $p$ values for GATA4 (DNase $p < 10^{-10}$, H3K27ac $p < 10^{-5}$) and GATA6 (DNase $p < 10^{-13}$, H3K27ac $p < 10^{-7}$). After this analysis is conducted for all TR ChIP-seq samples in the Cistrome DB and the results are combined and compared, GATA6 and GATA4 ChIP-seq from intestinal

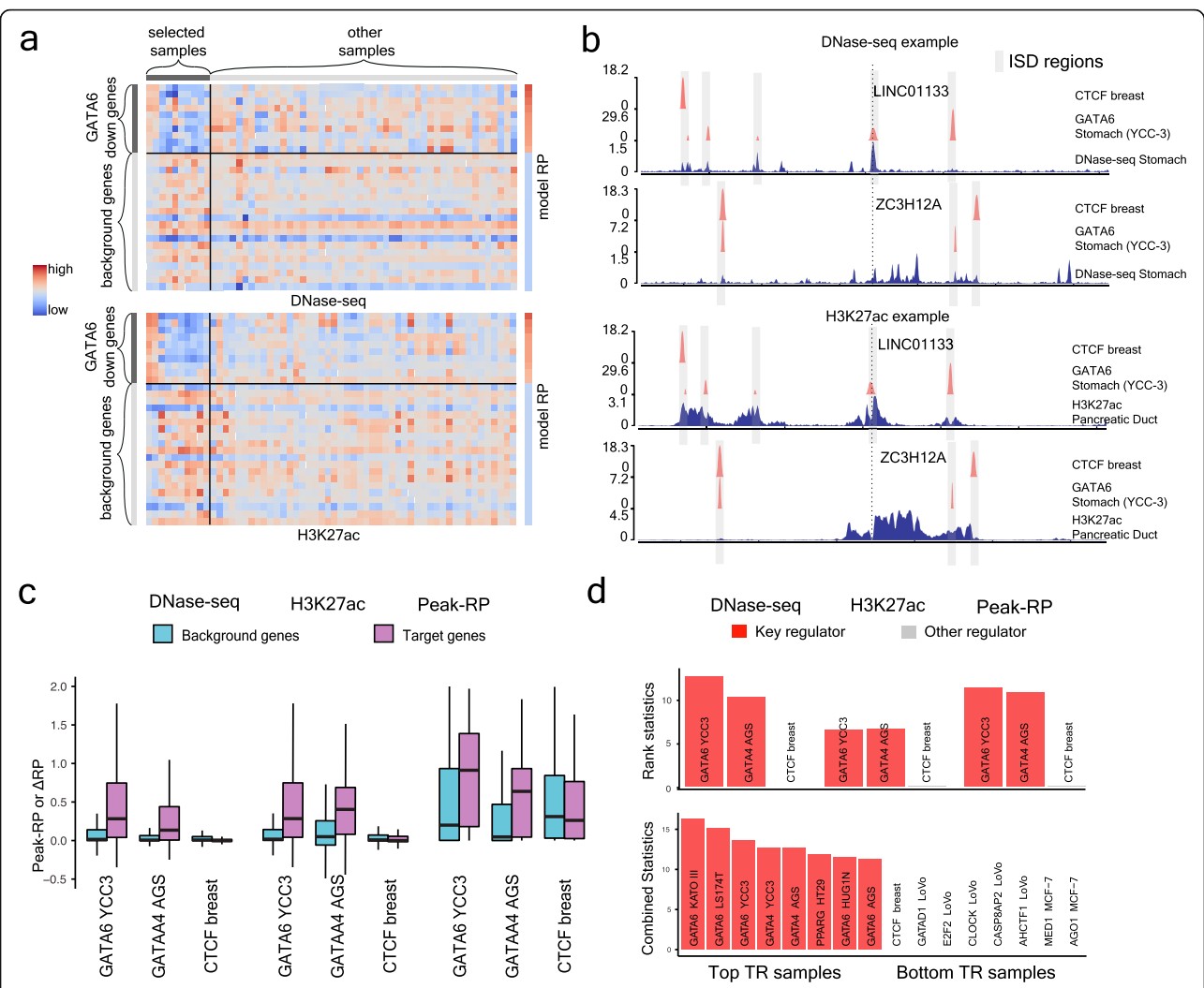

**Fig. 2** A downregulated gene set from a GATA6 knockdown experiment in gastric cancer KATO-III cells is used as a case study to demonstrate the Lisa framework. **a** Heatmap of regulatory potentials used to discriminate downregulated genes from non-regulated background genes. **b** In silico deletion analysis using GATA6 and CTCF cistromes to probe chromatin landscape models near an illustrative downregulated gene, LINC01133, and a background gene, ZC3H12A. Only the H3K27ac ChIP-seq and DNase-seq chromatin profiles with the largest positive coefficients are shown, although other samples contribute to the respective H3K27ac ChIP-seq and DNase-seq chromatin models. **c** Comparison of ΔRPs indicates GATA6 and GATA4 cistromes have a large impact on the chromatin landscapes near downregulated genes and are therefore likely to be regulators of the query gene set. CTCF does not influence the chromatin landscape of the downregulated genes and is not likely to regulate the query gene set. **d** The rank statistics for the Lisa analysis of the downregulated gene set in the GATA6 knockdown experiment were combined to get overall TR ranks. The top eight and the bottom eight TRs for all TR ChIP-seq samples are shown

and gastric tissues have the most significant *p* values (Fig. 2c, d).

## Lisa identification of regulatory TF ChIP-seq sample clusters

To investigate whether a TF ChIP-seq cistrome derived from one cell type can be informative about other cell types, we first clustered all the human TR cistromes in the Cistrome DB based on the pairwise Pearson correlation of peak-RP scores as a heatmap (Fig. 3). We then applied Lisa to differentially expressed gene sets defined by perturbations of individual TFs and examined the TR

cistromes predicted to be the key regulators of these gene sets. In the analysis of upregulated genes on androgen receptor (AR) activation in the LNCaP prostate cancer cell line, Lisa identified a tight cluster of significant cistromes for AR and its known collaborator FOXA1 (Fig. 3 (a)). All samples in this cluster were derived from prostate cancer cell lines. In the analysis of the GATA6 knockdown in the gastric cancer cell line (KATO-III), Lisa found the GATA6 and FOXA2 cistromes in the stomach and colon samples to be the most significant. FOXA2 is an important pioneer TF which has been reported to collaborate with GATA6 in gut development

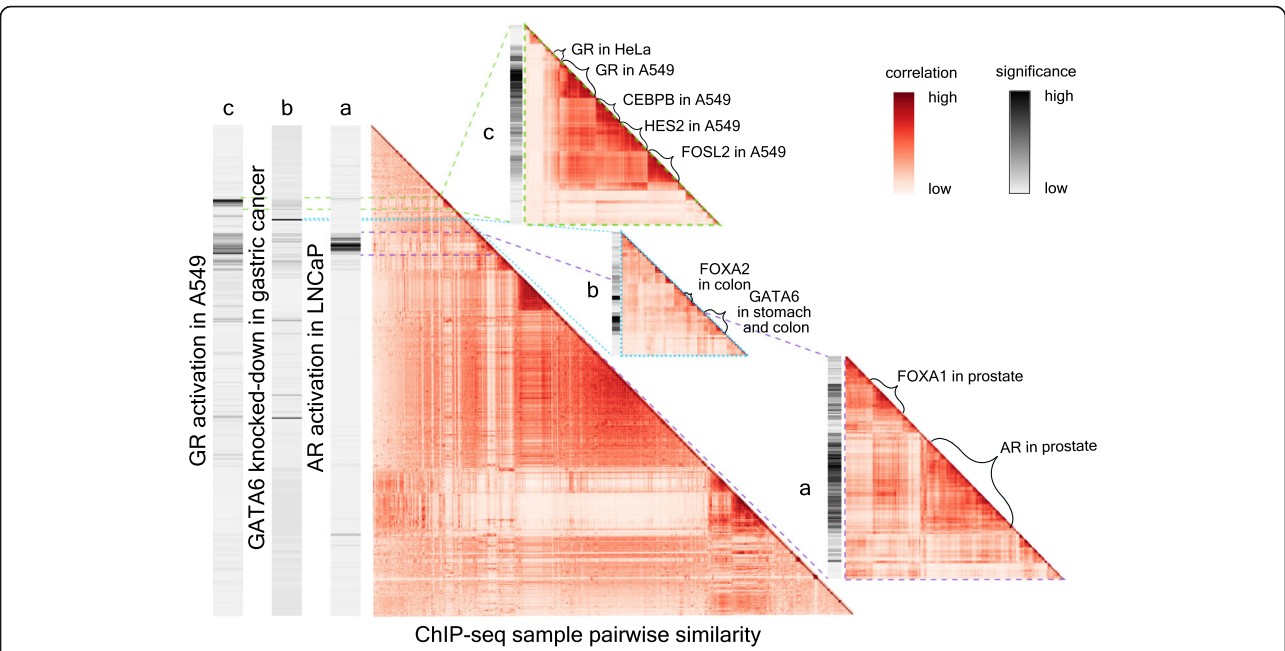

**Fig. 3** Lisa predicts key transcriptional regulators and assigns significance to each Cistrome DB cistrome. The large heatmap shows the hierarchical clustering of 8471 human Cistrome DB ChIP-seq cistromes based on peak-RP, with color representing Pearson correlation coefficients between peak-RPs. The three bars to the left of the heatmap display Lisa significance scores for differentially expressed genes sets derived from GR activation in the A549 cell line (upregulated), GATA6 knockdown in gastric cancer (downregulated), and AR activation in the LNCaP cell line (upregulated). Small heatmaps show details of the global heatmap relevant to (a) AR activation, (b) GATA6 knockdown, and (c) GR activation gene sets. In each case, the most significant cistromes are derived from the same cell type or lineage

to regulate Wnt6 [41] and Wnt7b [42] (Fig. 3 (b)). The identification of GATA6 cistromes in colon cancer cell lines, in addition to gastric cancer cell lines, shows that cistromes derived from cell types that are of related lineages can be used to inform the identification of the relevant regulators, even if the cell types are not the same. In the third example involving glucocorticoid receptor (GR) activation in the lung cancer cell line A549, Lisa correctly identified GR in A549 as a likely regulator and also identified GR in a different cell type HeLa (Fig. 3 (c)). AR, a member of the same nuclear receptor family as GR, is also implicated by Lisa even though the AR cistrome samples do not cluster with GR cistrome samples and have less statistical significance.

We carried out an analysis of the effects of removing ChIP-seq and DNase-seq data on Lisa's accuracy. In particular, we tested Lisa's performance on three upregulated gene sets: (1) GR-activated genes in breast cancer (MCF7), (2) GR-activated genes in lung cancer (A549), and (3) estrogen receptor (ER)-activated genes in MCF7 (Additional file 4: Table S3). In these analyses, we assessed the effect of removing all relevant cell-line-specific (MCF7 or A549), H3K27ac ChIP-seq and DNase-seq data, or cell-line-specific TR ChIP-seq data (ER or GR). We also removed cell-line-specific TR ChIP-seq data together with H3K27ac ChIP-seq and DNase-seq data. We repeated the same analysis removing similar

data, on the basis of tissue (breast and lung) instead of on the basis of cell line (MCF7 and A549). When MCF7 ER ChIP-seq are excluded, an ER sample from another breast cancer cell line (H3396) predicts the importance of ER (rank 6) as a regulator of the estrogen-activated gene set. When all ER breast ChIP-seq samples are excluded, Lisa can still identify ER (rank 18) from ER ChIP-seq in the VCaP prostate cancer cell line. For the GR-activated gene set in MCF7, when GR ChIP-seq data is unavailable in MCF7, Lisa can identify GR as a key regulator (rank 2) using GR ChIP-seq from the lung (A549). For the GR-activated gene set in the lung, Lisa identified GR as the key regulator (rank 1) using GR ChIP-seq data from the breast (MDA-MB-231). Together, these observations indicate that although TRs often bind in cell-type-specific ways, ChIP-seq-derived TR cistromes can be informative about the gene sets that TRs regulate in some other cell types.

## Lisa identification of TF-associated cofactors in addition to TFs

To illustrate Lisa's capacity to find cofactors that interact with the regulatory TFs, we examined the Lisa analyses of four differentially expressed gene sets derived from experiments involving the activation of GR [43] and the knockdown/out of BCL6 [44], MYC [45], and SOX2 [46]. Lisa analysis of GR activation in

lung cancer ranked GR itself as the most significant TR for the upregulated gene sets (Fig. 4a) and highly ranked pioneer TFs FOSL2 and CEBPB, which were downregulated after GR activation (Fig. 3c). BCL6, a predominantly repressive TF, is a driver of diffuse large B cell lymphoma (DLBCL) [47]. Lisa analysis of the upregulated genes in a BCL6 knockdown experiment in a DLBCL cell line ranked BCL6 as the most significant TR for this gene set (Fig. 4b). Lisa also identified NCOR1 and NCOR2, which are transcriptional BCL6 corepressors involved in the regulation of the germinal center [48–50]. SPI1, which recruits BCL6 [51], and BCOR, another BCL6 corepressor [52], were ranked among the top TRs for the upregulated gene set. In a MYC knockdown experiment in medulloblastoma, MYC and its dimerization partner, MAX [53], were among the top predicted regulators of the downregulated genes (Fig. 4c). The histone methyltransferase, KDM2B, known to physically interact with MYC and to augment MYC-regulated transcription [54] was also detected among the top regulators. In the SOX2 knockout experiment [2], NANOG, SOX2, and POU5F1, the key regulators of

pluripotency, were the top predicted regulators of the downregulated genes (Fig. 4d). Lisa also discovered a similar set of TRs for the gene set derived from a POU5F1 knockdown experiment in embryonic stem cells (Additional file 1: Figure S3,4a). In addition, β-catenin (CTNNB1), which interacts with SOX2 and is oncogenic in SOX2+ cells [55], also ranked high for the downregulated genes. The predicted regulators of the upregulated genes in this experiment include FOXA1 and EOMES. FOXA1 is involved in early embryonic development [56] and has been observed to repress NANOG directly [57]. FOXA1 has been shown through co-immunoprecipitation to physically interact with SOX2 [58]. SOX2, known to bind to an enhancer regulating EOMES in human ESCs, when knocked down triggers EOMES expression and induces endoderm and trophectoderm differentiation [59]. Thus, in many cases, the known interactors are highly ranked along with the target activator or repressor. This suggests that even though the available TF ChIP-seq data in different cell types are sparse (Additional file 1: Figure S1d), Lisa can provide insights on possible regulatory TFs since transcriptional

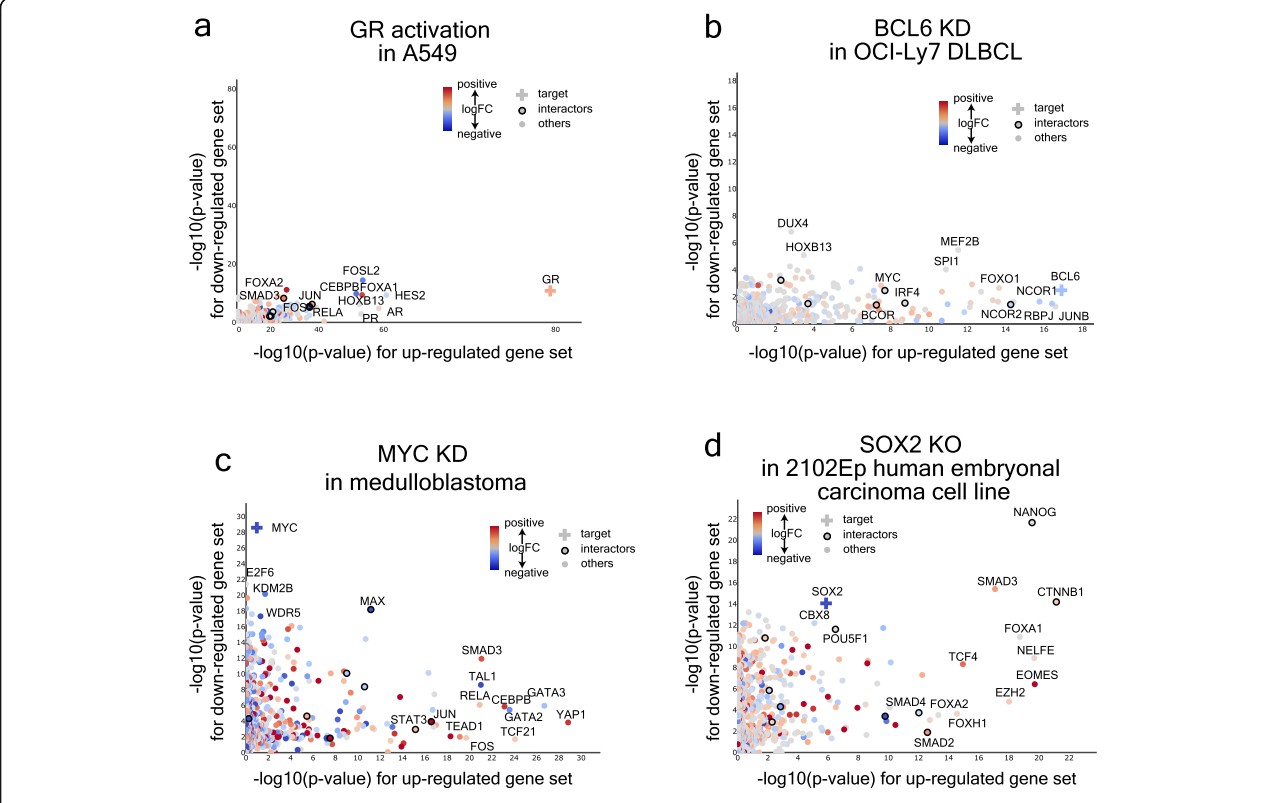

**Fig. 4** Lisa can accurately identify key transcriptional regulators and co-regulators using Cistrome DB cistromes. Lisa analyses of up- and downregulated gene sets from **a** GR overexpression, **b** BCL6 knockdown, **c** MYC knockdown, and **d** SOX2 knockout experiments. The scatter plots show negative $\log_{10}$ Lisa $p$ values of 1316 unique transcriptional regulators for up- and downregulated gene sets. Colors indicate $\log_2$ fold changes of the TF gene expression between treatment and control conditions in the gene expression experiments. Dots outlined with a circle denote transcriptional regulators that physically interact with the TF perturbed in the experiment, which is marked with a cross

machinery tends to be organized in modules of inter-
acting factors [60] (Additional file 1: Figure S4d).

### Systematic evaluation of regulator prediction

To systematically evaluate Lisa, we compiled a bench-
mark panel of 122 differentially expressed gene sets from
61 studies involving the knockdown, knockout, activa-
tion, or overexpression of 27 unique human target TFs.
In addition, we compiled 112 differentially expressed
gene sets derived from 56 studies with 25 unique TF
perturbations in mice (Additional file 5: Table S4, see
"galleries" at http://lisa.cistrome.org). The full Lisa
model was separately applied to the upregulated and
downregulated gene sets in each experiment. We also
carried out analyses of these gene sets using subcompon-
ents of Lisa: the peak-RP method, as well as H3K27ac
ChIP-seq- and DNase-seq-assisted ISD analyses. The pu-
tative regulatory cistromes were defined using either

ChIP-seq peaks or from TF motif occurrence in the in-
ferred chromatin models. The results allowed us to com-
pare the effectiveness of DNase-seq and H3K27ac ChIP-
seq in scenarios where the TF cistromes are well esti-
mated (by ChIP-seq) or less well estimated (by motif).
We measured the performance based on their ranking of
the perturbed target TF (Fig. 5, Additional file 1: Figure
S5).

We compared the performance of methods that use
TF ChIP-seq data and TF motifs, on up- and downregu-
lated gene sets, and on overexpression/activation and
knockdown/knockout samples (Fig. 5a). In overexpres-
sion studies, the prediction performance of all methods
tended to be better for the upregulated gene sets than
for the downregulated ones. The reverse is evident in
the knockout and knockdown studies for which the pre-
diction performances are better for the downregulated
gene sets (Fig. 5b, c). This suggests that most of the TFs

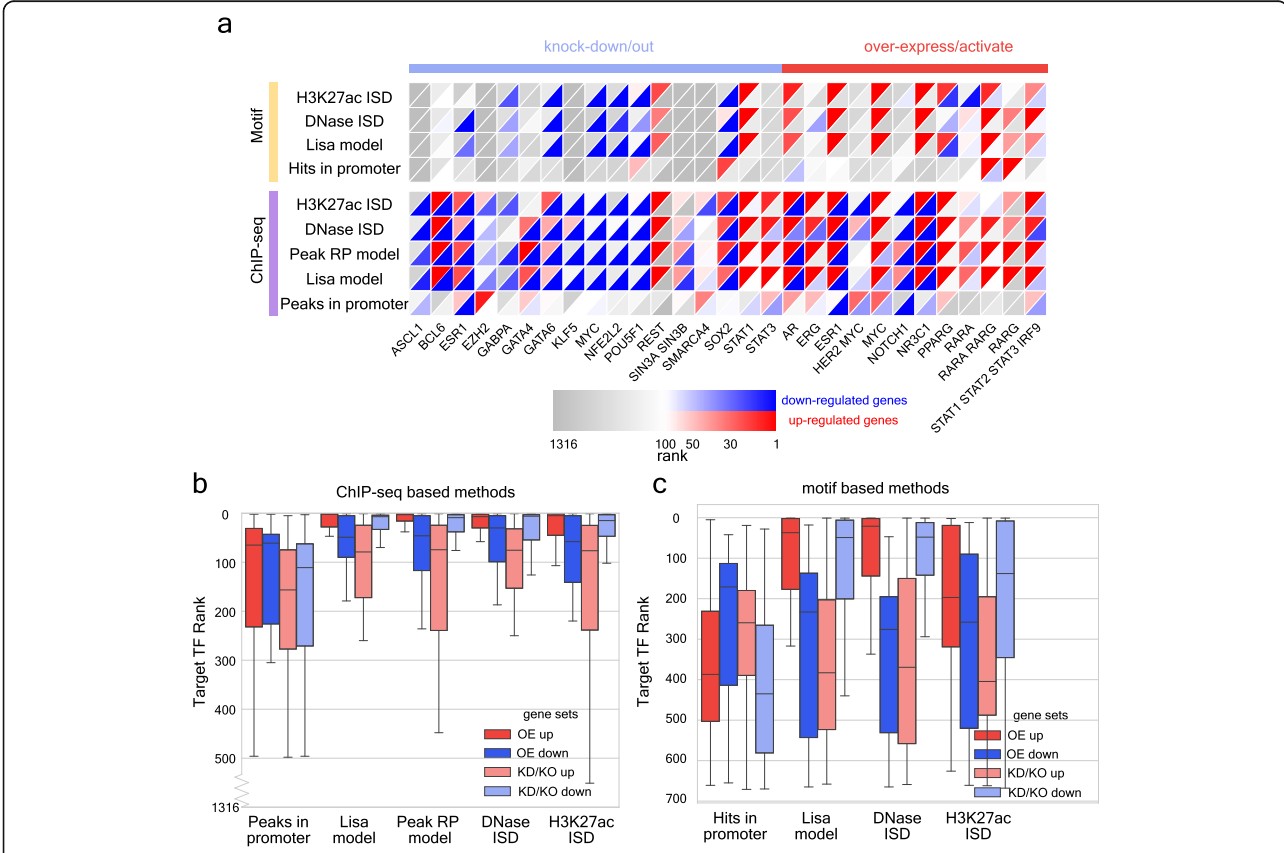

**Fig. 5** Systematic evaluation of regulator prediction performance for humans using Cistrome DB ChIP-seq and DNA motif-derived cistromes. **a**
Heatmap showing Lisa's performance in the analysis of human TF perturbation experiments. Each column represents a TF activation/
overexpression or knockdown/out experiment with similar experiment types grouped together. Rows represent the methods based on cistromes
from TR ChIP-seq data or imputed from motifs. The upper left red triangles represent the rank of the target TFs based on the analysis of the
upregulated gene sets; the lower right blue triangles represent the analysis of downregulated gene sets. The heatmap includes non-redundant
human experiments for the same TF. See Additional file 1: Figure S5 for the complete list of human and mouse experiments. **b** Boxplot showing
the target TF rankings comparing Lisa ChIP-seq-based methods and the baseline model based on TF peak counts in gene promoter regions to
analyze up- and downregulated gene sets in overexpression/activation (OE) and knockdown/out experiments (KD/KO). **c** Boxplot showing target
TF rankings using Lisa motif-based methods and the baseline model based on motif hits in promoter regions

included in the study have a predominant activating role in the regulation of their target genes, under the conditions of the gene expression experiments, allowing these TFs to be more readily identified with the corresponding direction of primary gene expression response. Similar performance patterns were observed in the mouse benchmark datasets (Additional file 1: Figure S5). The performances of Lisa using ISD of TR ChIP-seq peak from chromatin landscapes were similar to the TR ChIP-seq peak-RP method, but outperformed motif-based methods by large margins.

To determine whether differences between the up- and downregulated gene sets could be explained by direct or indirect modes of TR recruitment, we studied two experiments involving ER and GR activation in greater detail. We defined "direct" ER and GR binding sites as ER/GR ChIP-seq peaks on genomic intervals containing the cognate DNA sequence elements and "indirect" ER and GR binding sites as ER/GR ChIP-seq peaks without the sequence elements. Comparing direct and indirect binding sites in the respective ER and GR activation experiments (Additional file 1: Figure S6), we found that the upregulated gene sets were more significantly associated with the direct binding sites (ER $p$ value $1.5 \times 10^{-15}$, GR $p$ value $1.5 \times 10^{-18}$) than with the indirect ones (ER $p$ value $3.8 \times 10^{-4}$, GR $p$ value $1.4 \times 10^{-12}$). The downregulated gene sets were more significantly associated with the indirect binding sites (ER $p$ value $1.5 \times 10^{-15}$, GR $p$ value $1.5 \times 10^{-11}$) than with the direct ones (ER $p$ value $4.6 \times 10^{-2}$, GR $p$ value $3.0 \times 10^{-3}$).

In some cases, the perturbation of a TR may trigger stress, immune, or cell cycle checkpoint responses that are not directly related to the initial perturbation. In the Lisa analysis of upregulated genes after 24 h of estradiol stimulation (GSE26834), for example, E2F4 is the top-ranked TR, followed by ER. Estrogen is known to stimulate the proliferation of breast cancer cells via a pathway involving E2F4, a key regulator of the G1/S cell cycle checkpoint [61]. In this case, Lisa might be correctly detecting a secondary response to the primary TR perturbation.

## Comparison of Lisa with published methods

We next compared Lisa with other approaches, including BART [29], i-cisTarget [30], and Enrichr [31], which can use either TR ChIP-seq data or motifs. We also included a baseline method that ranks TRs by comparing query and background gene sets based on the TR binding site number within 5 kb centered on the TSS. Lisa outperformed BART, i-cisTarget, and Enrichr in terms of the percentage of the target TR identified within the top ten across all the experiments, either using TF binding sites from ChIP-seq data or motif hits (Fig. 6a, b). Lisa uses a model based on chromatin data to give more weight to the loci that are more likely to influence the expression of the query gene set. In this way, Lisa improves the performance of TR inference with noisy cistrome profiles such as those imputed from DNA sequence motifs. In addition to being more accurate than other methods in terms of TR prediction, the Lisa web server (lisa.cistrome.org) has several unique features that allow investigators to explore relevant ChIP-seq data in ways that are not available in other applications.

## Lisa web site and gallery of Lisa's benchmark data

The Lisa web site (lisa.cistrome.org) displays two tables of results for each query gene set. The first summarizes the Lisa analysis based on TR ChIP-seq data, and the second displays the Lisa analysis of TF binding sites imputed from DNA binding motifs. The ChIP-seq data

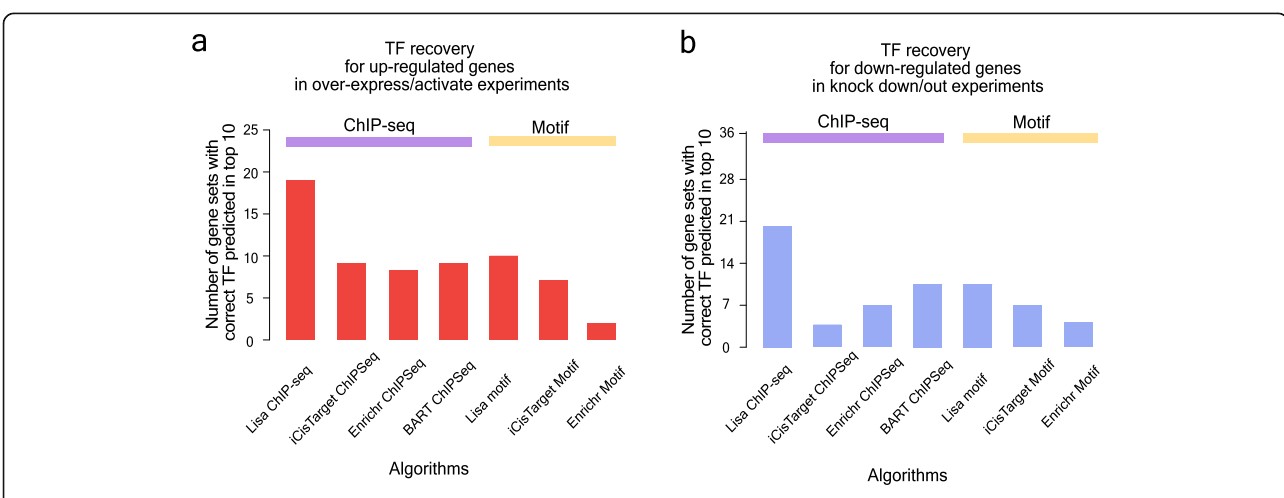

**Fig. 6** Lisa's performance surpasses published models. Lisa's performance is compared with alternative published methods for **a** upregulated genes in overexpression/activation experiments and **b** downregulated genes in knockdown/out experiments

table displays up to five ChIP-seq samples for each TR. Users can sort results by $p$ value and inspect metadata and quality control statistics for each of the ChIP-seq samples to understand whether the predictive samples may be derived from particular cell types or experimental conditions. Lisa provides quality control metrics, metadata, publication, and read data repository links for the ChIP-seq data of putative regulatory TRs. Through Lisa, the ChIP-seq signal tracks can be viewed on the WashU Epigenome Browser [62]. Although the motif imputation-based analysis tends to be less accurate than the ChIP-seq based analysis, motifs can indicate roles for regulatory TRs for which ChIP-seq data is not widely available. Lisa's analysis of all the benchmark gene sets is also viewable on the Lisa web site. Users can explore these analyses to understand the "typical" results of the analysis. Robust methods combined with visualization and data exploration features make Lisa a valuable tool for analyzing gene regulation in humans and mice.

## Conclusion

In this study, we describe an approach for using publicly available ChIP-seq and DNase-seq data to identify the regulators of differentially expressed gene sets in humans and mice. On the basis of a series of benchmarks, we demonstrate the effectiveness of our method and report recurrent patterns in the TRs predicted by these methods. We find the regulators of the upregulated genes and the downregulated ones are often different from each other; therefore, in any analysis of differential gene expression, up- and downregulated gene sets ought to be distinguished. Our results show that many TFs have a preferred directionality of effect, indicative of a predominant repressive or activating function. It is well known that many TFs can recruit both activating and repressive complexes [63], so the observed direction may be related to the stoichiometry and affinity of the activating or repressive cofactors. We also observe differences between ChIP-seq-based analysis and motif-based ones, suggesting differences in the TF activity depending on whether a TF interacts directly with DNA or whether it is recruited via another TF [64]. When a TF is recruited by another TF, it is likely that the enhancer has been already established by other TFs and protein complexes. Thus, the co-binding enhancer information of multiple TFs allows Lisa to identify both the DNA-bound TFs and their partners which might not directly bind DNA.

Lisa's accuracy in predicting the regulatory TRs of a gene set depends on the perturbation used in the production of the differential gene expression data; the quality of the gene expression data; the availability and quality of the DNase-seq, H3K27ac, and TR ChIP-seq data sets; the degree to which binding is dependent on a DNA sequence motif; and the validity of the model assumptions. Although we evaluate Lisa using differential gene expression data associated with a TR perturbation, the perturbed TR might not be the main regulator of the gene set. For example, perturbation of a TR may trigger a stress response [65] or secondary transcriptional effects that are not directly related to the primary TR [66].

The modeling approach used in Lisa facilitates the prediction of regulatory TRs using available ChIP-seq and DNase-seq data. DNase-seq and H3K27ac ChIP-seq are available in a broad variety of cell types, and these data are informative about *cis*-regulatory events mediated by many TRs. Although H3K27ac is considered to be a histone modification associated with gene activation, Lisa can still identify TRs, such as BCL6 and EZH2, with predominantly repressive functions. Although Lisa uses the correlation between H3K27ac or chromatin accessibility and gene expression to predict regulatory TRs, we do not assume that H3K27ac or chromatin accessibility causes the transcriptional changes. Other genomics data types that are predictive of general *cis*-regulatory activity, when available in quantity, variety, and quality, might improve Lisa's performance. More importantly, high-quality TR-specific binding data, generated by ChIP-seq or alternative technologies, like CETCh-seq [14] or CUT & RUN [15], will be needed to improve Lisa's accuracy in predicting TRs that are not yet well represented in Cistrome DB. TR imputation methods might fill in some gaps in TR binding data; however, families of TRs such as homeobox and forkhead factors, which have similar DNA-binding motifs, can be hard to discriminate based on DNA sequence analysis.

Although Lisa aims to identify the regulators of any differentially expressed gene set in humans or mice, no matter the contrast, in practice, the query gene sets should be derived from biologically meaningful differential expression or co-regulation analyses. In this study, we based the method evaluation on data from available TR perturbation experiments, which are biased towards well-studied systems. For this reason, the reliability of methods based on TR ChIP-seq data may be overestimated relative to imputation-based methods because the available TR ChIP-seq data tends to be derived from similar cell types and for the same factors used in the gene perturbation experiments. When the relevant cell-type-specific TR ChIP-seq data is available, the performance of the peak-RP method and ISD methods are similar, but when TR ChIP-seq data is not available, methods based on imputed TR cistromes are obligatory. The value of imputed cistromes relative to ChIP-seq derived ones will depend on the quantity, variety, and quality of available ChIP-seq data; the accuracy of the imputed cistromes; the degree of commonality of the genes that are regulated by the same TR in different cell

types; and the number of TRs recognizing similar DNA sequence elements. Lisa provides invaluable information about the regulation of gene sets derived from both bulk and single-cell expression profiles [67] and will become more accurate over time with greater coverage of TF ChIP-seq augmented by computationally imputed TF cistromes.

## Methods

### Preprocessing of chromatin profiles

Using the BigWig format signal tracks of human and mouse H3K27ac ChIP-seq and DNase-seq from Cistrome DB, we precomputed the chromatin profile regulatory potential (chrom-RP) of each RefSeq gene and also summarized the signal in 1-kb windows genome-wide. The chrom-RP for gene $k$ in sample $j$ is defined as $R_{jk} = \sum_{i \in [t_k - L, t_k + L]} w_i s_{ji}$ (as defined in the MARGE algorithm [28]). $L$ is set to 100 kb, and $w_i$ is a weight representing the regulatory influence of a locus at position $i$ on the TSS of gene $k$ at genomic position $t_k$, $w_i = 2e^{-\mu d}/1 + e^{-\mu d}$, where $d = |i - t_k|/L$, and $i$ stands for $i$th nucleotide position within the $[-L, L]$ genomic interval centered on the TSS at $t_k$. $s_{ji}$ is the signal of chromatin profile $j$ at position $i$. $\mu$ is the parameter to determine the decay rate of the weight, which is defined as $\mu = -\ln L \Big/ 3\Delta$. For DNase-seq and H3K27ac ChIP-seq, the decay distance $\Delta$ is set to 10 kb. The genome-wide read counts on 1-kb windows were calculated using the UCSC utility bigWigAverageOverBed [68]. The chrom-RP matrix for chromatin profiles was normalized across RefSeq genes within one chromatin profile by $R'_{jk} = \log(R_{jk} + 1) - \frac{1}{k} \sum_1^k (\log(R_{jk} + 1))$.

### Preprocessing of cistromes

We converted TR ChIP-seq peaks from the Cistrome Data Browser (v.1) BED files into binary values to represent binding within 100bp resolution genomic intervals. DNA sequence scores were calculated from Cistrome DB position weight matrices, a redundant collection of 1,061 PWMs from TRANSFAC [69], JASPAR [70] and Cistrome DB ChIP-seq, representing 675 unique TFs in human and mouse. The peak-based regulatory potential (peak-RP) of a TR cistrome is defined in the same way as the chrom-RP except $s_i$ represents the presence ($s_{ji} = 1$) or absence ($s_{ji} = 0$) of a peak summit within the upstream and downstream 100 kb centered on TSS. The genome-wide motif scores were scanned at a 100-bp window size with the library (https://github.com/qinqian/seqpos2) [71], and the motif hits are defined by thresholding at the 99th percentiles then mapped to the 1-kb windows. The genome-wide 1-kb

windows in which the TR peak summits are located were determined using Bedtools [72]. All of the peak-RPs, TR binding, and motif hit data were deposited into hdf5 format files.

### Lisa framework

#### Chromatin landscape model

Lisa selects 3000 background genes by proportionally sampling from non-query genes with a range of different TAD and promoter activities based on compendia of Cistrome DB H3K4me3 and H3K27ac ChIP-seq signals. There is no gene ontology enrichment in the background gene set. Lisa then uses L1-regularized logistic regression to select an optimum sample set for H3K27ac ChIP-seq or DNase-seq samples based on $R'_{jk}$. The L1 penalty parameter is determined by binary search to constrain the number of selected chromatin profiles to be small but sufficient to capture the information (different sample sizes were explored, and 10 was used in all the benchmark cases [28]). Lisa trains a final logistic regression model to predict the target gene set and obtains a weight $\alpha_j$ for each candidate chromatin profile $j$, from which the weighted sum of chrom-RP is the model regulatory potential (model-RP).

#### In silico deletion method

The rationale for the ISD method is that the peaks of the true regulatory TFs should align with the high chromatin accessibility signals from the corresponding tissue or cell type. Therefore, the computational deletion of the chromatin signals on the peaks of regulatory cistromes would result in a more substantial effect on the model-RP for query genes than for background genes. The regulatory potentials are recalculated after erasing the signal in all 1-kb windows containing at least one peak from a putative regulatory cistrome $i$, $\tilde{R}_{ijk} = R_{jk} - \sum_{m \in M_{ik}} l\, w_m s_{jm}$ (where $M_{ik}$ is the set of 1-kb windows containing at least one peak in cistrome $i$ for gene $k$; $l$ is the window size, which is set to 1 kb for this study; $w_m$ is the exponential decay weight with the distance between the $m$th window center and TSS, the weight function is the same as chrom-RP; and $s_{jm}$ is the $j$th average chromatin profile signal on the $m$th window). These RPs are then normalized using the same normalization factors from the original RPs $\tilde{R}'_{ijk} = \log(\tilde{R}_{ijk} + 1) - \frac{1}{K} \sum_1^K (\log(R_{jk} + 1))$.

After deletion, the model RPs are recalculated using the weights from the logistic regression model from chromatin profile feature selection without refitting and subtracted from the non-deletion model-RP, producing a ΔRP value for each gene, defined as the linear

combination of differences in regulatory potentials: $\Delta R'_{ik} = \sum_j \alpha_j (R'_{jk} - \tilde{R}'_{ijk})$.

### Combined statistics method for TR ranking

The peak-RPs or ΔRPs of the query gene set are compared with that of the background gene set through the one-sided Wilcoxon rank-sum test. For ChIP-seq-based methods, peak-RP, DNase-seq, and H3K27ac chom-RP are combined to get a robust prediction of the TRs. For motif-based methods, DNase-seq and H3K27ac ΔRPs are combined to get the final inference of TRs. Both combinations of statistics follow the Cauchy combination test [38], in which the combined statistics for each TR is $t_j = \sum_{i=1}^{d} w_i \tan\{(0.5-p_i)\pi\}$, where $j$ represents one TR, $i$ represents the $i$th method within ChIP-seq-based or motif-based methods, $p_i$ is the corresponding $p$ value, and $w_i$ is set to $1/d$ where $d$ is 3 for ChIP-seq-based method or 2 for the motif-based method. The combined $p$ value for a TR $j$ is computed as $p_j = 1/2 - (\arctan(t_j))/\pi$.

### Baseline method

The baseline method, which is the "peaks in promoter" for ChIP-seq-based method or "hits in promoter" for the motif-based method, is implemented by counting the number of TF ChIP-seq binding summits or motif hits within the genomic interval from 5 kb upstream to 5 kb downstream of the TSS. The peaks or motif counts in the promoter of the target gene set are compared with that of the background gene set using the one-sided Wilcoxon rank-sum test.

### Comparison of "direct" and "indirect" binding sites

For up- and downregulated gene sets from the same experiment, the peaks of the target TR ChIP-seq samples with the most significant $p$ values are defined as "direct" or "indirect" binding sites based on the target TR motif scores. Peak-RPs of "direct" or "indirect" binding sites are calculated and normalized to percentiles. Statistical significance between query and background gene sets was calculated by the one-sided Wilcoxon rank-sum test.

### Comparison of Lisa with published methods

All up- and downregulated gene sets in Lisa's benchmark dataset were also used to test other published methods. BART and i-cisTarget were manually run through online websites with the default settings. Enrichr was run using the API. When comparing the motif-based methods, PWMs from species other than humans or mice were removed since they are not included in the Lisa framework: BART (http://bartweb.

org/), i-cisTarget (https://gbiomed.kuleuven.be/apps/lcb/i-cisTarget/?ref=labworm), and Enrichr (http://amp.pharm.mssm.edu/Enrichr/).

### Lisa pipeline

The Lisa pipeline is implemented with Snakemake [73]. Lisa contains an interface to process FASTQ format files to BigWig format files and to generate hdf5 files containing the chrom-RP matrices and 1-kb resolution data required by the Lisa model module.

### Lisa online application

We have implemented the online version of Lisa (http://lisa.cistrome.org) using the Flask Python web development framework, along with process control software Celery to queue numerous queries. The analysis result of the target gene set is closely linked to the Cistrome DB. The scatterplot comparing TR ranking results from a pair of query gene sets such as up- and downregulated gene sets are implemented in Plot.ly.

## Supplementary information

---

**Additional file 1.** Figure S1-S6.

**Additional file 2:** Table S1. Cistrome profile annotation table including TR ChIP-seq and TF motifs.

**Additional file 3:** Table S2. DNase-seq and H3K27ac sample annotation table for mouse and human.

**Additional file 4:** Table S3. Analysis of Lisa predictions of GR and ER regulated genes using data which does not match the specific cell type. The cell line and cell type of the highest ranked Lisa predicted target TR sample are shown in parentheses in each case.

**Additional file 5:** Table S4. TF perturbation DNA microarray meta table for benchmarking the peak-RP and Lisa methods.

**Additional file 6:** Review history.

---

**Peer review information**

**Review history**
The review history is available as Additional file 6.

**Authors' contributions**
CM, XSL, and MB conceived the concept and initiated the project. CM, QQ, and JF developed the Lisa algorithm. QQ implemented the Lisa software and website. JF and QQ evaluated Lisa's performance and carried out the analysis of the results. JF collected and processed the gene expression data. JF, QQ, RZ, CW, QW, and HS collected and processed the ChIP-seq and DNase-seq data. CM, XSL, and JZ supervised the project. CM, XSL, QQ, JF, and JZ wrote the paper. All authors read and approved the final manuscript.

**Funding**
This work was supported by grants from the NIH (U24 HG009446 to XLS, U24 CA237617 to XSL and CM), National Natural Science Foundation of China (31801110 to SM), and the Shanghai Sailing Program (18YF1402500 to QQ).

## Availability of data and materials

Lisa is available under the MIT open source license at https://github.com/liulab-dfci/lisa [74] and at Zenodo [75]. All TR ChIP-seq, DNase-seq, and H3K27ac ChIP-seq data are from the Cistrome Data Browser (http://cistrome.org/db) [71]. Gene expression profiles used for benchmarking the method were accessed at Gene Expression Omnibus (https://www.ncbi.nlm.nih.gov/geo/). The lists of all the data used in this study are available in the additional files. The processed gene lists and Lisa results are available at the gallery of the Lisa server (http://lisa.cistrome.org).

## Ethics approval and consent to participate

Not applicable.

## Competing interests

MB receives sponsored research support from Novartis. MB serves on the SAB of Kronos Bio and is a consultant to H3 Biomedicine. XSL is a cofounder and board member of GV20 Oncotherapy, SAB of 3DMed Care, consultant for Genentech, and stockholder of BMY, TMO, WBA, ABT, ABBV, and JNJ. All other authors declare that they have no competing interests.

## Author details

[1]Clinical Translational Research Center, Shanghai Pulmonary Hospital, School of Life Science and Technology, Tongji University, Shanghai 200433, China. [2]Center of Molecular Medicine, Children's Hospital of Fudan University, Shanghai 201102, China. [3]Stem Cell Translational Research Center, Tongji Hospital, School of Life Science and Technology, Tongji University, Shanghai 200065, China. [4]Center for Functional Cancer Epigenetics, Dana-Farber Cancer Institute, Boston, MA 02215, USA. [5]Department of Medical Oncology, Dana-Farber Cancer Institute, Harvard Medical School, Boston, MA 02215, USA. [6]Department of Data Sciences, Dana-Farber Cancer Institute and Harvard T.H. Chan School of Public Health, Boston, MA 02215, USA.

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

## 

