## [**Additional file 6:** Review history. · Genome Biology]

Review History

First round of review

Reviewer 1

Were you able to assess all statistics in the manuscript, including the appropriateness of statistical tests used? Yes.

Were you able to directly test the methods? Yes.

Comments to author:

Qin and colleagues report a new method, Lisa, that predicts transcriptional regulators of query gene sets versus a background gene set based on inferred regulatory importance derived from H3K27ac ChIP-seq and DNase-seq data. Lisa is an extension of previous efforts from the group, that used inferred regulatory importance from peaks. The strategy seems promising overall and is of general interest to the community. However, there are some clarifications and demonstrations needed, which I list below.

* The introduction and the first sections (including the section on identification of sample clusters) would really benefit from a rewrite to make the overall approach and rationale more comprehensible.

* In particular, I lack a schematic overview and text of the Lisa approach that guides the reader through the rationale of the method, and how this compares to Peak-RP.

* There are several aspects of the Lisa approach that is not justified or, if justified elsewhere, not cited. These include the normalization approach and the assumptions of the decay model and additivity of multiple binding sites. How do these assumptions affect the final results? Can additivity be justified for all loci? How does that fit with models of regulatory redundancy? The authors need to motivate these choices or discuss the potential impact these design choices may have.

* My biggest concern with the approach is the sparsity of available data. The authors indeed point out that improvement and new insights will lead from new available data. However, a demonstration of the influence of data availability is lacking. How are results affected if data would be reduced to even more sparsity for certain samples?

Reviewer 2

Were you able to assess all statistics in the manuscript, including the appropriateness of statistical tests used? Yes: Statistics used appear appropriate.

Were you able to directly test the methods? Yes.

Comments to author:

Review:

The authors describe Lisa, an analysis framework that takes a set of regulated genes from a user and then interrogates a database of DNase/ChIP-seq experiments to identify key regulators likely to play a role in regulating the given genes. Lisa works by first selecting a limited set of H3K27ac and DNase experiments from the Cistrome DB that contain 'regulatory potentials' that discriminate the input genes relative to a set of background genes, then performs an in silico perturbation of these profiles using ChIP-seq profiles for TFs and other co-factors from Cistrome DB to identify which factors are predicted to influence the H3K27ac/DNase patterns the most in the vicinity of target genes (based on their RP model). To validate their model, the authors analyze a compendium of knockdown/out and overexpression experiments for

specific regulators with the expectation that the perturbed factor should be recovered by Lisa. They also compare how well Lisa identifies the correct factor compared to the performance of i-CisTarget and Enricher.

This work attempts to address a pressing need in the field of gene regulation by attempting to leverage the incredible amount of chromatin profiling information available (i.e. Cistrome DB) to make accurate predictions about which transcription factors or other regulators are likely to play roles in regulating specific sets of genes. This work builds on previous efforts (MARGE/BART) by including the interrogation of TF/co-factor ChIP-seq profiles and refining how they are used to predict the most significant regulators. Although this work is somewhat incremental, for most part the method is well described and should be a useful tool for genomics researchers.

Major Comments:

One important section missing from this manuscript is a more complete description of what Lisa provides as results and a discussion of how to interpret them. This is important for how readers will use and interpret the information given by Lisa. It seems the primary output of Lisa is a list of TRs and p-values (i.e. visualized in Fig. 4). Most of the results in the manuscript stress that Lisa works because the expected CR appears near the 'top' of the results. What other TRs appear significant for typical experiments? Fig. 3 attempts to address this by commenting that significant TRs often include master TFs found in the cell types (or related cell types) from which the gene lists were derived. Are there other trends a user should be aware of? Are there certain types of experiments for which Lisa performs worse than others? For example, in the list of experiments analyzed in Fig. 5 to evaluate performance, if other factors were ranked above the expected TR, what were they? Is there an explanation/reason for it, and if so, could this be problematic for analyzing certain types of data?

Related to the above comment: Due to the selection of RP based on H3K27ac and DNase, TR profiles generated in the same or related cell types as the 'selected' H3K27ac/DNase experiments would be expected to gravitate toward the top of the results. This is not a problem per se, since they are more likely to be important, but it is somewhat a trivial result. It would be interesting to see how Lisa scores the TRs from the same cell type relative to one another. For example, when analyzing E2 regulated genes in MCF-7 cells, I would imagine Lisa would predict ERa, FoxA1, etc. are significant - but how does Lisa prioritize these (and other TRs) from MCF7 cells relative to one another? Can it discriminate TRs that drive key aspects of gene regulation versus TRs responsible for general cell lineage specification or simply accumulate in regions of open chromatin?

The authors make several claims/observations about how Lisa doesn't need profiles (TR or H3K27ac/DNase) from the systems being studied to make inferences about key TRs (i.e. they comment on how analyzing NR3C1 activation in A549 cells identifies NR3C1 from a Hela experiment). It would be nice to test this explicitly at both the H3K27ac/DNase and TR level so that users know how well this can be expected to work for their systems. For example, if analyzing MCF7 estrogen regulated genes, what would happen if you remove all of the H3K27ac/DNase breast cancer samples from the analysis such that Lisa will need to use other profiles? Or instead, what if the breast cancer TR profiles (i.e. TFs/co-factors) are removed - how well will Lisa perform if only TRs from other cell types are used? (or what if both K27ac/DNase and TRs for breast cancer cells are removed and the prediction must be made using data only from other cell types). In general, it is important to understand how holes in database coverage impact the Lisa's results.

Minor comments:

Related to the comment above about database coverage, the authors include a nice control addressing how their model predicts factors based on DNA motifs instead of using TR profiles. First, it is unclear how the authors deal with TF motifs vs. TF motif families in their reporting/scoring of accuracy - since many TFs bind essentially the same motif within their family, scoring the accuracy of this step could be trickier and may require special consideration (not really explained at all in the methods or addressed in the manuscript). Second, it may be interesting to compare the motif version to a result where the TRs for the cell type of interest are withheld - comparing how well motif inference based on sequence compares to TF inference based on TR profiles from other cell types.

In the discussion, the authors comment on direct vs. indirect binding of TFs based on motifs, but don't really offer a direct assessment of how that information may (or may not) be that useful. Is it possible to segregate TR profiles for TFs based on peaks with or without the predicted motif to see how these different sets of sites perform in their model?

The introduction could use a couple statements about the challenges in modeling changes in gene expression based on chromatin/TFs profiles. The idea of using K27ac/DNase and creating an additive model with exponential decay etc. to describing gene regulation and the pros and cons of this approach (or alternatives) are not really discussed. Given this is one of the most important assumptions that goes into this work it would be nice to address this.

RP (regulatory potential) could be added to the list of abbreviations since it's a crucial aspect of the models described.

The comparison between LISA, BART and MARGE is mentioned in the conclusion but not available or displayed in this manuscript. This would be nice to include (I realize Fig. 5 may contain a version of this, but it would be nice to show this comparison more explicitly)

Reviewer 3

Were you able to assess all statistics in the manuscript, including the appropriateness of statistical tests used? Yes. No additional statistical review is recommended.

Were you able to directly test the methods? No.

Comments to author:

In this manuscript, Qin et al describe a method called Lisa to 'identify' (more like predict) transcriptional regulators (TRS) of a query gene set (differentially expressed or co-expressed) using a precompiled collection of publicly available data on chromatin accessibility (H3K27ac ChIP-Seq and DNase-Seq) and TR binding sites (inferred from ChIP-Seq peaks and/or motif hits based on PWMs from TRANSFAC and JASPAR databases), which the authors have compiled in their cistrome data browser (Zheng et al., NAR 2019 & Mei et al., NAR 2017). Lisa, which can be broken down into four basic steps, is quite straightforward, focusing on the +/-5 Kb region immediately up and downstream of gene promoters.

The first step involves selection of a background (control) gene set for the query set of genes. This is done by selecting 3000 genes by proportionally sampling non-query genes with different promoter activities (based on H3K4me3 and H3K27ac signals, associated with active gene promoters) and TADs. The authors state that "there is no GO enrichment in the background gene set." It is not clear how they ensure this unless they repeatedly select a background gene set and run GO enrichment analysis until the selected set of background genes are not enriched for any specific GO term (the authors should clarify this point).

Using a strategy similar to the one the authors employed in their previous method MARGE-express (ref. 27; Wang et al., 2016 Genome Research), they use logistic regression to select an optimum sample set for H3K27ac ChIP-Seq and DNase-Seq samples; as in their previous paper, they settle on ten for their sample size (small enough size to capture the information).

The second step involves computation of regulatory potential of chromatin profile (chrom-RP) using the ten H3K27ac and DNase-Seq datasets identified in step one, again using the strategy the authors used previously in another of their methods called MARGE-potential (ref. 27; Wang et al., 2016 Genome Research paper). Lisa then goes on to use these chromatin signals to train a L1-regularized logistic regression model that discriminates the query gene set from background gene set, and uses the weighted sum of chrom-RP to arrive at what is called a model regulatory potential (model-RP).

The third step involves computation of regulatory potential of ChIP-Seq peaks (peak-RP) of TRs, very similar to the way they compute chrom-RP (ref. 27).

In the last step, the authors look for cistromes that produce higher peak-RP values for the query gene set than for the background gene set (one-sided Wilcoxon rank-sum test is used as the test statistic). TRs with the most significant p-values are considered to be the candidate TRs of the query gene set. And, each of the candidate TR is evaluated by estimating the effect deleting each TR cistrome has on the chromatin landscape model. This is done by setting DNase-Seq and H3K27ac ChIP-Seq signal to zero within 1 Kb regions containing TR peaks and evaluating the predicted effect on the model-RP. The difference between the model scores before and after ISD (delta-RP) is used to assess the impact the 'deleted' TR cistrome is predicted to have on the query and background gene sets, followed by ranking of the candidate TRs.

The authors go on to show Lisa's utility through case studies on a few gene sets (from KD/KO or overexpression experiments) and make the case that they are able to recover the TR (KD/KO/OE) that can best explain the gene sets. Lisa's performance using TR ChIP-Seq peaks or TF motifs was systematically evaluated by applying Lisa on 100+ differentially expressed gene sets of >50 mouse and human studies. Lisa was applied separately on up- and down-regulated genes in each experiment, and the prediction performance for TF ChIP-Seq data-based method is better compared to TF motif-based method. Finally, Lisa is compared to two other methods iCisTarget and Enricher, and Lisa is shown to do well against the two.

While there is no doubt that Lisa would be a nice addition to the set of available tools for predicting/inferring transcription regulators, methodologically I do not see any novel in particular other than recycling of their previously published strategies. Specifically, while the strategies used in the first three steps are conceptually no different from the authors' previous methods, the last step is a variation of their MARGE-potential (ref. 27). Whereas MARGE-potential, which does not use TR ChIP-Seq datasets, computes the regulatory potential of cis-regulatory environment surrounding the TSS, Lisa by way using TR ChIP-Seq datasets is able to predict TRs. In some ways, Lisa is nothing more than a reincarnation of MARGE, which the authors correctly acknowledge as "the second descendent of MARGE".

Major points:

1. The peak-RP model the authors use assumes that the effect a TF binding site has on the expression of a

gene decays exponentially with genomic distance between the TF binding site and TSS. A hallmark of cis-regulatory elements (enhancers) that has been repeatedly demonstrated is that they are relatively insensitive to distance or position relation to their target genes (Shlyueva et al., 2014). And, proximity of a TF binding site to a gene has been shown to be a poor predictor of target genes. Given these facts, I wonder how reasonable the authors assumption is given that it may not have any basis. Given that we now have Hi-C type of maps for many cell types, wouldn't using such information be better for predicting TRs?

2. I appreciate the authors' sentiment that ChIP-Seq data accurately characterize genome-wide TF binding sites, but what the authors fail to take into account or acknowledge is that not all TF binding sites are functional. Numerous studies have shown that hundreds if not thousands of genes, with a TF binding site (ChIP-Seq) at their promoter, exhibit no change in expression when the TF is knocked out/down or over-expressed. Given this, assuming that all ChIP-Seq peaks, and thus inferred TF binding sites, are functional may not be the way to go.

3. The authors also seem to make the assumption that all TRs are activators of gene expression, which we all know is not true. Many TFs have been shown to both activate and repress gene expression. And, there are several chromatin regulators that are negative regulators of gene expression. This assumption and associated bias may be why the authors find that (lines 242-251): "In over-expression studies, the prediction performance of all methods tended to be better for the up-regulated gene sets, than for the down-regulated gene ones. The reverse is evident [true] for the knock-out and knock-down studies for which the prediction performances are better for the down-regulated gene sets (Fig. 5b,c). This suggest that most of the TFs included in the study have a predominant activating role in the regulation of their target genes."

4. Also, recently research has shown that transcription at enhancers located within genes can also repress the expression of their host gene. How would this affect the proposed model?

5. Lines 100-104 & 365-366: Not sure if one-sided Wilcoxon rank sum test is appropriate here because it assumes that TRs are activators and that the computational deletion of TF cistrome is expected to reduce the model score. Two-sided Wilcoxon rank sum test should be used instead.

6. Line 201: "Lisa analysis of the up-regulated genes in a BCL6 knockout experiment in a DLBCL cell line ranked BCL6 first (Fig. 4b)." Explain how BCL6 is an up-regulated gene in a BCL6 knockout experiment. Shouldn't BCL6 be down-regulated in a BCL6 knockout experiment?

7. Lisa does well on query gene sets (Fig. 4) that are more cell type-specific and/or targets of cell type-specific TRs. The authors should test Lisa on more difficult query gene sets, which are targets of generic TRs that are ubiquitously expressed. This would widen the method's appeal to those who study basic biological processes (that are not just associated with development, organ/tissue, or diseases in particular).

Minor Points:

1. I recommend using 'predict' instead of 'identify' in the abstract (line 21) and in the main text.

2. Line 59-60: "ChIP-Seq data availability is limited and many cistromes have not been produced for important TFs in many cell types (ENCODE consortium paper, 2012)". This statement was probably true back in 2012, but given the explosion of ChIP-Seq datasets, the authors may want to tone it down. Availability of antibodies with greater specificity is the limiting factor to ChIP-Seq data availability these days, which should be rightfully acknowledged.

3. Line 67, [promoters of] actively transcribed genes

4. Lines 163-169 of the section "Demonstration of Lisa on a Gata6 knockdown study": Statements containing 'small delta-RP', 'little difference', 'most significant p-values' all need to be backed up p-values (which is missing). And, results from the analyses should be provided as a supplementary table so readers can evaluate the performance of Lisa

5. The data used to generate Fig. 3 heatmaps should be provided as supplementary (excel/data) tables

6. While the main text says 'BCL6 knockout,' figure 4b title says "BCL6 KD": which one is it?

Inferring transcriptional regulators through integrative modeling of public chromatin accessibility and ChIP-seq data

Qian Qin; Jingyu Fan; Rongbin Zheng; Shenglin Mei; Changxin Wan; Qiu Wu; Hanfei Sun; Jing Zhang; Myles Brown; Clifford Meyer; X. Shirley Liu

GBIO-D-19-00911: Response to Reviewers' Comments

We thank the reviewers for their time and effort in reviewing this manuscript and believe that their commentary has enabled us to substantially improve the revised version. Below we provide a point by point response to each of the reviewers' comments. The reviewers' comments are in black regular font followed by our responses in *blue italics*.

Reviewer #1: Qin and colleagues report a new method, Lisa, that predicts transcriptional regulators of query gene sets versus a background gene set based on inferred regulatory importance derived from H3K27ac ChIP-seq and DNase-seq data. Lisa is an extension of previous efforts from the group, that used inferred regulatory importance from peaks. The strategy seems promising overall and is of general interest to the community. However, there are some clarifications and demonstrations needed, which I list below.

We appreciate the reviewer for seeing the promise and interest of Lisa to the community. We have found Lisa to be invaluable to several ongoing research projects in the lab, to identify transcription regulators (TRs) involved in resistance to cancer immune therapy, prostate cancer metastasis, cancer drug treatment response, single cell clusters, and E3 ubiquitin ligase substrates.

* The introduction and the first sections (including the section on identification of sample clusters) would really benefit from a rewrite to make the overall approach and rationale more comprehensible.

We have revised the description of Lisa and the schematic representation in Figure 1 to explain the approach and rationale more clearly.

* In particular, I lack a schematic overview and text of the Lisa approach that guides the reader through the rationale of the method, and how this compares to Peak-RP.

In the revision we have included a schematic in Figure 1, which shows the roles of the Peak-RP and in silico deletion models.

* There are several aspects of the Lisa approach that is not justified or, if justified elsewhere, not cited. These include the normalization approach and the assumptions of the decay model and additivity of multiple binding sites. How do these assumptions affect the final results? Can additivity be justified for all loci? How does that fit with models of regulatory redundancy? The authors need to

motivate these choices or discuss the potential impact these design choices may have.

In the revision we have included a section to motivate the approach and describe the assumptions of our model. While it is true that there can be redundancy of regulation where multiple TR binding sites regulate the same gene, some TR binding sites appear to not regulate nearby genes. As this is not well understood, system specific data is needed to probe the details of this phenomenon. With the available data and current understanding of gene regulation we cannot justify a model that more complex than an additive one. We have also discussed the difference between a quantitative model that can be used to predict TRs given differential gene sets and the scientific literature of findings on gene regulation that are mostly based on few genes in few cell types. While these studies are important for understanding the mechanisms of transcriptional regulation, many observations cannot be included in a general quantitative model because the data needed to model these effects is not available on a genome-wide scale across a large variety of cell types and conditions. The main factor limiting the power of our method, however, is the availability of TR binding data.

* My biggest concern with the approach is the sparsity of available data. The authors indeed point out that improvement and new insights will lead from new available data. However, a demonstration of the influence of data availability is lacking. How are results affected if data would be reduced to even more sparsity for certain samples?

*While it is true TRs ChIP-seq data is sparse, data available in related cell types can still be informative. This is one of the motivations for the ENCODE project, and ENCODE data has been widely used by the research community to model gene regulation in many other cell types, conditions, and disease (e.g. GWAS) settings. We carried out an analysis of data availability for the Estrogen and Glucocorticoid Receptors (ER and GR), which are well represented in several cell types (see Table R1). In this analysis, we show that in the absence of GR ChIP-seq data in MCF7 Lisa can still identify GR as the key regulator of GR activation in **breast** (MCF7), by using GR ChIP-seq in **lung** (A549). In the analysis of estrogen (E2) stimulated genes in **breast** (MCF7), when all ER ChIP-seq data in breast cell lines or tissues are excluded from the analysis, Lisa is still able to identify ER, although with slightly worse rank, using ER ChIP-seq data from the VCaP **prostate** cancer cell line. Lisa will increase in accuracy as data from individual labs as well as consortia such as ENCODE continue to generate more experimental or imputed data. We have included this analysis in the revised manuscript.*

Reviewer #2: Review:

The authors describe Lisa, an analysis framework that takes a set of regulated genes from a user and then interrogates a database of DNase/ChIP-seq

experiments to identify key regulators likely to play a role in regulating the given genes. Lisa works by first selecting a limited set of H3K27ac and DNase experiments from the Cistrome DB that contain 'regulatory potentials' that discriminate the input genes relative to a set of background genes, then performs an in silico perturbation of these profiles using ChIP-seq profiles for TFs and other co-factors from Cistrome DB to identify which factors are predicted to influence the H3K27ac/DNase patterns the most in the vicinity of target genes (based on their RP model). To validate their model, the authors analyze a compendium of knockdown/out and overexpression experiments for specific regulators with the expectation that the perturbed factor should be recovered by Lisa. They also compare how well Lisa identifies the correct factor compared to the performance of i-CisTarget and Enricher.

This work attempts to address a pressing need in the field of gene regulation by attempting to leverage the incredible amount of chromatin profiling information available (i.e. Cistrome DB) to make accurate predictions about which transcription factors or other regulators are likely to play roles in regulating specific sets of genes. This work builds on previous efforts (MARGE/BART) by including the interrogation of TF/co-factor ChIP-seq profiles and refining how they are used to predict the most significant regulators. Although this work is somewhat incremental, for most part the method is well described and should be a useful tool for genomics researchers.

We thank the reviewer for appreciating the potential value Lisa brings to the research community. We ourselves are using Lisa extensively to identify transcription regulators (TRs) involved in resistance to cancer immune therapy, prostate cancer metastasis, cancer drug treatment response, single cell clusters, and E3 ubiquitin ligase substrates. Lisa contains significant conceptual advances as well as practical improvements in gene regulation analysis that are completely absent from MARGE and BART. MARGE does not make any prediction of the TRs that regulate a gene set. BART was developed independently by Dr. Zang's group at the University of Virginia and uses a different approach to identify TRs. While BART analysis is based on an enrichment analysis of MARGE-predicted putative enhancers, Lisa uses a different, chromatin landscape model, and in silico deletion approach to analyze the likely effects of TRs on gene expression. As such, Lisa is a far more accurate predictor of regulatory TRs than BART (Fig R1). In addition, a significant amount of software engineering and implementation effort made Lisa into a resource truly useful to the community (lisa.cistrome.org).

Figure R1. Comparison of Lisa prediction performance with available TR prediction methods. Although Lisa and BART are related to MARGE, Lisa is far more accurate. (a) Prediction of up-regulated genes in TR over-expression experiments. (b) Prediction of down-regulated genes in TR knock-down/out experiments.

Major Comments:

One important section missing from this manuscript is a more complete description of what Lisa provides as results and a discussion of how to interpret them. This is important for how readers will use and interpret the information given by Lisa. It seems the primary output of Lisa is a list of TRs and p-values (i.e. visualized in Fig. 4). Most of the results in the manuscript stress that Lisa works because the expected CR appears near the 'top' of the results. What other TRs appear significant for typical experiments?

We have added a section describing the Lisa web site (lisa.cistrome.org) features and results. As results differ between gene sets, we show some different scenarios in the manuscript. To help users to further understand the specific association between the Lisa identified factors with their gene sets, we provide the Lisa analysis results on a large number of benchmarking gene sets: http://lisa.cistrome.org/new_gallery/new_gallery.html

Fig. 3 attempts to address this by commenting that significant TRs often include master TFs found in the cell types (or related cell types) from which the gene lists were derived. Are there other trends a user should be aware of? Are there certain types of experiments for which Lisa performs worse than others? For example, in the list of experiments analyzed in Fig. 5 to evaluate performance, if other factors were ranked above the expected TR, what were they? Is there an explanation/reason for it, and if so, could this be problematic for analyzing certain types of data?

The reviewer raises interesting and important questions. Several factors are likely to determine Lisa's performance. These include the specific perturbation condition for the differential gene expression data, the quality of the gene expression data, the availability and quality of the DNase-seq and H3K27ac and TR ChIP-seq data sets, and the degree to which binding is dependent on a DNA sequence motif. In some TR perturbation experiments there might be some TRs that have stronger effects on expression than the perturbed TR, which might be why Lisa ranked these higher than the presumed target TR. The perturbation of a TR may also trigger stress, immune or cell cycle responses that are not directly related to the initial perturbation. In the Lisa analysis of up-regulated genes after 24 hours of estradiol stimulation (GSE26834), E2F4 is the top ranked TR, followed by ESR1. Estrogen is known to stimulate proliferation of breast cancer cells via a pathway involving E2F4, a key regulator of the G1/S cell cycle checkpoint (Carroll et al, JBC, 2000). In this case, Lisa appears to be correctly detecting a secondary response to the initial TR perturbation. We discuss these points in the revised manuscript.

Related to the above comment: Due to the selection of RP based on H3K27ac and DNase, TR profiles generated in the same or related cell types as the 'selected' H3K27ac/DNase experiments would be expected to gravitate toward the top of the results. This is not a problem per se, since they are more likely to be important, but it is somewhat a trivial result.

When available, the expected chromatin profiles are often the most important components of the Lisa chromatin model, although this is not always the case. The Lisa in silico deletion method emphasizes the cis-elements that are likely to be important in the regulation of the gene set while deemphasizing the ones that are less important. The less important ones can be regions that are active in many cell types. This quantification is best done using a model that considers large numbers of samples. In other words, the model defines a contrast between chromatin profiles where background estimation is important in addition to foreground estimation. We have clarified this point in the revised manuscript.

It would be interesting to see how Lisa scores the TRs from the same cell type relative to one another. For example, when analyzing E2 regulated genes in MCF-7 cells, I would imagine Lisa would predict ERa, FoxA1, etc. are significant - but how does Lisa prioritize these (and other TRs) from MCF7 cells relative to one another?

Lisa predicts TRs that tend to bind in regions with relevant chromatin model properties close to the regulated genes. The top ranked TRs are not necessarily from the same cell type, although this is often the case. In the example of the ER activation in MCF7 cells (GSE26834), E2F4 samples from B-lymphocyte and retinal pigment cells are ranked higher than ER. We provide the complete results of the benchmark gene sets on the Lisa web site lisa.cistrome.org, to allow users to see details of each ChIP-seq data set. In cases where there are several

ChIP-seq data sets for the same TR in the same cell type investigators can even inspect details of specific experiments.

Can it discriminate TRs that drive key aspects of gene regulation versus TRs responsible for general cell lineage specification or simply accumulate in regions of open chromatin?

Lisa can discriminate TRs for general vs specific gene regulation, which is achieved through the user-provided differential expression gene sets. Lisa input gene sets can be derived from different perturbation experiments ranging from targeted TR perturbation to drug treatments and developmental studies, and in each of these cases the nature of the key driver TRs might be different. In some studies, it may be important to understand the pioneer TFs while in others the pioneers are of little interest. To add flexibility to Lisa we therefore allow users to define their own background gene sets that can be defined to better frame their question.

The authors make several claims/observations about how Lisa doesn't need profiles (TR or H3K27ac/DNase) from the systems being studied to make inferences about key TRs (i.e. they comment on how analyzing NR3C1 activation in A549 cells identifies NR3C1 from a HeLa experiment). It would be nice to test this explicitly at both the H3K27ac/DNase and TR level so that users know how well this can be expected to work for their systems. For example, if analyzing MCF7 estrogen regulated genes, what would happen if you remove all of the H3K27ac/DNase breast cancer samples from the analysis such that Lisa will need to use other profiles? Or instead, what if the breast cancer TR profiles (i.e. TFs/co-factors) are removed - how well will Lisa perform if only TRs from other cell types are used? (or what if both K27ac/DNase and TRs for breast cancer cells are removed and the prediction must be made using data only from other cell types). In general, it is important to understand how holes in database coverage impact the Lisa's results.

We thank the reviewer for this helpful suggestion to understand the impact of data coverage on prediction accuracy. We have carried out the suggested analyses and included the results in the revised manuscript. In particular, we tested Lisa's performance on three up-regulated gene sets: (1) Glucocorticoid Receptor (GR) activated genes in breast cancer (MCF7), (2) Glucocorticoid Receptor (GR) activated genes in lung cancer (A549), and (3) Estrogen Receptor (ER) activated genes in MCF7. In these analyses (Supplementary Table), we assessed the effect of removing all relevant cell line specific (MCF7 or A549), H3K27ac ChIP-seq and DNase-seq data, or cell line specific TR ChIP-seq data (ER or GR). We also removed cell line specific TR ChIP-seq data together with H3K27ac ChIP-seq and DNase-seq data. We repeated the same analysis removing similar data, on the basis of tissue (breast and lung) instead of on the basis of cell line (MCF7 and A549). When MCF7 ER ChIP-seq are excluded, an ER sample from another breast cancer cell line (H3396) predicts the importance

of ER (rank 6) as a regulator of the estrogen activated gene set. When all ER breast ChIP-seq samples are excluded, Lisa can still identify ER (rank 18) from ER ChIP-seq in the VCaP prostate cancer cell line. For the GR activated gene set in MCF7, when GR ChIP-seq data is unavailable in MCF7, Lisa can identify GR as a key regulator (rank 2) using GR ChIP-seq from lung (A549). For the GR activated gene set in lung, Lisa identified GR as the key regulator (rank 1) using GR ChIP-seq data from breast (MDA-MB-231).

Gene set	H3K27ac and DNase data	All TR ChIP-seq included	Cell line target TR data excluded	Cell type target TR data excluded
ER activation in MCF7	All data	1 (MCF7, breast)	6 (H3396, breast)	18 (VCaP, prostate)
	Breast excluded	1 (MCF7, breast)	6 (H3396, breast)	18 (VCaP, prostate)
GR activation in MCF7	All data	2 (A549, lung)	2 (A549, lung)	2 (A549, lung)
	Breast excluded	2 (A549, lung)	2 (A549, lung)	2 (A549, lung)
GR activation in A549	All data	1 (A549, lung)	1 (MDA-MB-231, breast)	1 (MDA-MB-231, breast)
	Lung excluded	1 (A549, lung)	1 (MDA-MB-231, breast)	1 (MDA-MB-231, breast)

Table R1. Analysis of Lisa predictions using data which does not match the specific cell type.

Minor comments:

Related to the comment above about database coverage, the authors include a nice control addressing how their model predicts factors based on DNA motifs instead of using TR profiles. First, it is unclear now the authors deal with TF motifs vs. TF motif families in their reporting/scoring of accuracy - since many TFs bind essentially the same motif within their family, scoring the accuracy of this step could be trickier and may require special consideration (not really explained at all in the methods or addressed in the manuscript). Second, it may be interesting to compare the motif version to a result where the TRs for the cell type of interest are withheld - comparing how well motif inference based on sequence compares to TF inference based on TR profiles from other cell types.

Lisa currently does not take TF motif families into consideration, the statistical tests were taken independently for each motif position weight matrix (pwm) and deduplicated by the TR name associated with the (pwm). The detailed information of each motif is listed in Supplementary Table 1. Although TR ChIP-seq data appear to be more informative than motifs, meaningful quantification of motif based binding site imputation versus ChIP-seq is challenging for several reasons. First, some TRs bind in cell type specific ways whereas others have a more constant binding pattern, so the utility of TR binding data will be dependent on the nature of the TR. Second, the utility of ChIP-seq data in a different cell type will depend on the quality of the data and how related the cell types are. Third, the binding of some TRs, such as pioneer TFs, is more dependent on motifs than others, especially cofactors. Finally, this analysis could only answer how well our current motif approach would work, not what the full potential of

motif-based methods is. Machine learning methods for TR binding site imputation, such as those proposed in a recent ENCODE-DREAM challenge (Li et al, Genome Research, 2019), can make much better predictions than more naïve methods, but these methods have yet to be deployed on a large scale in practical applications. In the revised manuscript, we have added more details to the discussion of this issue.

In the discussion, the authors comment on direct vs. indirect binding of TFs based on motifs, but don't really offer a direct assessment of how that information may (or may not) be that useful. Is it possible to segregate TR profiles for TFs based on peaks with or without the predicted motif to see how these different sets of sites perform in their model?

In the manuscript we noted that the motif-based analyses tend to be more predictive in one direction of response than the other, while ChIP-seq based analyses tend to be able to predict well in both directions. We carried out analyses of several TF data sets, including GR activation in A549 cells (Fig. R2), to characterize this phenomenon, finding that direct binding sites do tend to behave differently from indirect ones. We have included these analyses in the revised manuscript.

Figure R2. Comparison of direct binding sites (GR peaks with motif) with indirect binding sites (GR peaks without motif). The direct sites are more predictive of GR activated genes than of GR repressed genes.

The introduction could use a couple statements about the challenges in modeling changes in gene expression based on chromatin/TFs profiles. The idea of using K27ac/DNase and creating an additive model with exponential decay etc. to describing gene regulation and the pros and cons of this approach (or alternatives) are not really discussed. Given this is one of the most important assumptions that goes into this work it would be nice to address this.

We have discussed more of the model assumptions and considerations in the introduction.

RP (regulatory potential) could be added to the list of abbreviations since it's a crucial aspect of the models described.

We have added RP to the list of abbreviations.

The comparison between LISA, BART and MARGE is mentioned in the conclusion but not available or displayed in this manuscript. This would be nice to include (I realize Fig. 5 may contain a version of this, but it would be nice to show

this comparison more explicitly)

We did not report a comparison with BART in the original submission because the BART web server was too slow to run all the test data. Recently BART has been updated to run faster. Our analysis shows Lisa to perform much better than BART (Fig. R1) and we have included this result in the revision. MARGE does not implement the TR prediction function so we cannot compare Lisa to MARGE, only to BART.

Reviewer #3: In this manuscript, Qin et al describe a method called Lisa to 'identify' (more like predict) transcriptional regulators (TRS) of a query gene set (differentially expressed or co-expressed) using a precompiled collection of publicly available data on chromatin accessibility (H3K27ac ChIP-Seq and DNase-Seq) and TR binding sites (inferred from ChIP-Seq peaks and/or motif hits based on PWMs from TRANSFAC and JASPAR databases), which the authors have compiled in their cistrome data browser (Zheng et al., NAR 2019 & Mei et al., NAR 2017). Lisa, which can be broken down into four basic steps, is quite straightforward, focusing on the +/-5 Kb region immediately up and downstream of gene promoters.

Lisa does not focus on a +/-5kb region, instead it uses a regulatory potential model that considers binding up to 100kb from the transcription start site (TSS), weighted as a function of genomic distance from the TSS. We have revised the description of Lisa and Figure 1 to explain the method more clearly.

The first step involves selection of a background (control) gene set for the query set of genes. This is done by selecting 3000 genes by proportionally sampling non-query genes with different promoter activities (based on H3K4me3 and H3K27ac signals, associated with active gene promoters) and TADs. The authors state that "there is no GO enrichment in the background gene set." It is not clear how they ensure this unless they repeatedly select a background gene set and run GO enrichment analysis until the selected set of background genes are not enriched for any specific GO term (the authors should clarify this point).

A randomly selected set of 3000 background genes, if correctly implemented, is not expected to be enriched in any GO term.

Using a strategy similar to the one the authors employed in their previous method MARGE-express (ref. 27; Wang et al., 2016 Genome Research), they use logistic regression to select an optimum sample set for H3K27ac ChIP-Seq and DNase-Seq samples; as in their previous paper, they settle on ten for their sample size (small enough size to capture the information).

The second step involves computation of regulatory potential of chromatin profile (chrom-RP) using the ten H3K27ac and DNase-Seq datasets identified in step one, again using the strategy the authors used previously in another of their

methods called MARGE-potential (ref. 27; Wang et al., 2016 Genome Research paper). Lisa then goes on to use these chromatin signals to train a L1-regularized logistic regression model that discriminates the query gene set from background gene set, and uses the weighted sum of chrom-RP to arrive at what is called a model regulatory potential (model-RP).

The third step involves computation of regulatory potential of ChIP-Seq peaks (peak-RP) of TRs, very similar to the way they compute chrom-RP (ref. 27). In the last step, the authors look for cistromes that produce higher peak-RP values for the query gene set than for the background gene set (one-sided Wilcoxon rank-sum test is used as the test statistic). TRs with the most significant p-values are considered to be the candidate TRs of the query gene set. And, each of the candidate TR is evaluated by estimating the effect deleting each TR cistrome has on the chromatin landscape model. This is done by setting DNase-Seq and H3K27ac ChIP-Seq signal to zero within 1 Kb regions containing TR peaks and evaluating the predicted effect on the model-RP. The difference between the model scores before and after ISD (delta-RP) is used to assess the impact the 'deleted' TR cistrome is predicted to have on the query and background gene sets, followed by ranking of the candidate TRs.

Lisa calculates three measures of association between a TR and the gene set based on H3K27ac delta-RP, DNase-seq delta-RP, and peak-RP. The results are integrated using the Cauchy Combination test. We have clarified this process in the revised manuscript.

The authors go on to show Lisa's utility through case studies on a few gene sets (from KD/KO or overexpression experiments) and make the case that they are able to recover the TR (KD/KO/OE) that can best explain the gene sets. Lisa's performance using TR ChIP-Seq peaks or TF motifs was systematically evaluated by applying Lisa on 100+ differentially expressed gene sets of >50 mouse and human studies. Lisa was applied separately on up- and down-regulated genes in each experiment, and the prediction performance for TF ChIP-Seq data-based method is better compared to TF motif-based method. Finally, Lisa is compared to two other methods iCisTarget and Enricher, and Lisa is shown to do well against the two.

While there is no doubt that Lisa would be a nice addition to the set of available tools for predicting/infering transcription regulators, methodologically I do not see any novel in particular other than recycling of their previously published strategies. Specifically, while the strategies used in the first three steps are conceptually no different from the authors' previous methods, the last step is a variation of their MARGE-potential (ref. 27).

Lisa is substantially different from MARGE and BART. In the schematic Figure 1c in the revised manuscript, the only part of Lisa that is similar to MARGE and BART relates to yellow circles 1 and 2 in the H3K27ac section.

Whereas MARGE-potential, which does not use TR ChIP-Seq datasets, computes the regulatory potential of cis-regulatory environment surrounding the TSS, Lisa by way using TR ChIP-Seq datasets is able to predict TRs. In some ways, Lisa is nothing more than a reincarnation of MARGE, which the authors correctly acknowledge as "the second descendent of MARGE".

The development of Lisa involved a significant amount of innovation and effort to produce a tool that would be useful for the research community. Our goal was to develop an effective system that can determine the TR regulators of a gene set, a function that MARGE did not perform at all. BART, makes this prediction yet we show that Lisa's performance is far better than BART's (Fig. R1), despite sharing some conceptual similarities. In addition, Lisa involves important original concepts. In particular, Lisa does not carry out a direct analysis of the genomic regions inferred by MARGE, which is BART's strategy. Lisa probes chromatin profile models to test the relative effects different TRs have on gene expression. In the revision we have improved Figure 1 to clarify the description of the Lisa method. While TR ChIP-seq is scarce for most factors H3K27ac ChIP-seq and DNase-seq data are available in hundreds of cell types, and these data can help to determine where the enhancers and promoters are likely to be in the cell types relevant to the query gene set. The chromatin profile models discriminate the differentially expressed genes from the background on the basis of H3K27ac or DNase regulatory potentials, which are weighted sums of ChIP-seq or DNase-seq reads. Lisa tests the effects of TRs by subtracting reads close to the TR binding sites and quantifying the effect of these local subtractions on the overall chromatin profile model. The difference in performance between Lisa and BART clearly demonstrates that what might seem to be minor methodological considerations are actually important.

Major points:

1. The peak-RP model the authors use assumes that the effect a TF binding site has on the expression of a gene decays exponentially with genomic distance between the TF binding site and TSS. A hallmark of cis-regulatory elements (enhancers) that has been repeatedly demonstrated is that that they are relatively insensitive to distance or position relation to their target genes (Shlyueva et al., 2014). And, proximity of a TF binding site to a gene has been shown to be a poor predictor of target genes.

The regulatory potential model assumes that the influence of a TF decreases monotonically as a function of distance between the TSS and the TF binding site. In genomics studies distance thresholds are often used to associate ChIP-seq peaks with genes. A distance threshold approach would be the same as using a step decay function in which all peaks inside the threshold are as likely to contribute to the regulation of the gene. If the threshold were 100kb, a peak at the transcription start site would have the same regulatory effect as a peak 100kb

from the TSS. We consider a smooth decay function to be a more reasonable model than a sharp distance cutoff. Physical laws describing the frequencies of interactions between components of a polymer or molecular interactions such as Van der Waals forces or dipole-dipole interactions are governed by smooth functions of distance. Distance is also important in models of physical processes such as gas diffusion or heat transfer. It can be clearly seen in Hi-C chromatin interaction data, that chromatin interaction frequency is highly correlated with genomic proximity.

Regarding enhancers being “relatively insensitive to distance in relation to their target genes”, we would like to bring a historical perspective. When enhancers were discovered “the prevailing view at the time that eukaryotic genes were controlled by promoters with a local influence, limited to about 100bp from the initiation site” (from a historical perspective by Schaffner was, *Biol. Chem.*, 2015). At the time, it was surprising that cis-regulatory elements could regulate genes across distances greater than 1kb: “these results meant nothing less than that the SV40 ‘enhancing’ DNA segment was able to boost transcription independent of its orientation and at distances of more than 1000 bp from a (related or unrelated) target promoter! And it even worked from a position downstream of the transcription unit. These properties were subsequently generally accepted as an enhancer definition” (Schaffner, *Biol. Chem.*, 2015). Clearly, enhancers can influence genes across distances far greater than 1kb, and they are **not** completely insensitive to distance. In fact, there is ample evidence, as exemplified below, showing that cis-elements near the transcription start site of a gene are **more likely** to regulated the gene than elements further away. We are **not** saying that there are no instances of more distal cis-elements having a greater regulatory influence than more proximal ones. This is a statement of probability, and the purpose of our regulatory potential model is quantify this probability.

Visel et al, 2009 Fig 5a. p300 peaks are enriched near genes that are expressed in the same tissue.

Consider, for example, a p300 ChIP-seq study of tissue specific enhancers in which proximity to the regulated gene is shown to be a good predictor of the regulation of tissue specific genes (Visel et al, *Nature*, 2009, Fig 5a). In this Figure, the blue bars are forebrain derived p300 peaks close to genes that are upregulated in forebrain, and the grey are random sites. The enrichment of tissue

specific enhancers relative to tissue specific genes decays with distance and approaches random background at around 100kb.

Or consider an analysis of GWAS catalogue variants that shows a strong relationship between eQTL density and genomic distance to the transcription start site of the regulated genes. This study found eQTL-target gene density to decrease with the distance between eQTLs and their cognate gene TSSs. This is strong evidence that enhancer regulation of genes follows a probabilistic function relative to the distance between enhancers and gene TSSs (Fig. 2d, Gamazon et al, Nat. Genetics, 2018).

Gamazon et al, Nat. Genetics, 2018 Fig 2d.

Additivity of the influence of multiple binding sites is consistent with each of the binding sites influencing the gene independently. The development of models of synergistic or antagonistic effects between enhancers would require experimental data, such as combinatorial cis-element CRISPR knockouts, that is unavailable on a genome wide scale. Therefore, we use a model of the H3K27ac environment surrounding each gene and the in silico deletion approach to capture information about whether a TR binding site is likely to be of regulatory relevance.

Given these facts, I wonder how reasonable the authors assumption is given that it may not have any basis. Given that we now have Hi-C type of maps for many cell types, wouldn't using such information be better for predicting TRs?

The 4D Nucleome data portal currently reports a total of 12 human Hi-C and 17 human in situ Hi-C data sets. If Hi-C data represents cell type and condition specific chromatin interactions, this small number does not seem to be a promising basis for a TR prediction system. Besides, the value of Hi-C data for enhancer inference is questionable as findings of several recent chromatin interaction papers suggest a rather limited role of stable chromatin loops in gene regulation (El Khattabi et al, 2019; Alexander et al, eLife, 2019; Benabdallah, Cell, 2019). The role of TADs in gene regulation has also been called into question (Ghavi-Helm, Nature Genetics, 2019). This quote from a review paper in Cell (Long et al. Cell, 2016) sums up the lack of census regarding the value of Hi-C: "What is also becoming clear is that, while there are certainly well documented examples of enhancer-promoter loops, typical enhancer-promoter contacts are likely less stable and/or less frequent than structural loops mediated by CTCF that are readily detectable by Hi-C methods." We have demonstrated an effective method for determining regulators without the use of Hi-C data that is limited in number and value.

2. I appreciate the authors' sentiment that ChIP-Seq data accurately characterize genome-wide TF binding sites, but what the authors fail to take into account or acknowledge is that not all TF binding sites are functional. Numerous studies

have shown that hundreds if not thousands of genes, with a TF binding site (ChIP-Seq) at their promoter, exhibit no change in expression when the TF is knocked out/down or over-expressed. Given this, assuming that all ChIP-Seq peaks, and thus inferred TF binding sites, are functional may not be the way to go.

We do not assume that all TR ChIP-seq peaks are functional but rather that each TR ChIP-seq peak has some probability of influencing nearby genes. It has been shown that the H3K27ac marked enhancer is indicative of active enhancers. The H3K27ac chromatin landscape modeling might help to discriminate between active enhancers and inactive ones. In the revision we clarify that not all TR binding sites are functional and that we take this into account in our model although with current data we cannot accurately predict which are functional and which are not.

3. The authors also seem to make the assumption that all TRs are activators of gene expression, which we all know is not true. Many TFs have been shown to both activate and repress gene expression. And, there are several chromatin regulators that are negative regulators of gene expression. This assumption and associated bias may be why the authors find that (lines 242-251): "In over-expression studies, the prediction performance of all methods tended to be better for the up-regulated gene sets, than for the down-regulated gene ones. The reverse is evident [true] for the knock-out and knock-down studies for which the prediction performances are better for the down-regulated gene sets (Fig. 5b,c). This suggest that most of the TFs included in the study have a predominant activating role in the regulation of their target genes."

Lisa does not assume that all TRs are activators. The method analyzes the association between gene sets and TR ChIP-seq peaks and the signs of the coefficients in the chrom-RP model can be negative or positive. If the query gene set is up-regulated in response to the inhibition of a predominantly repressive TR, as in the case of BCL6, Lisa could identify such a TR, as it does in the case of BCL6. What we observed in Lisa's analysis of differential gene expression data is that the directionality of effect for many TFs appears to be more consistent with an activating effect.

4. Also, recently research has shown that transcription at enhancers located within genes can also repress the expression of their host gene. How would this affect the proposed model?

It is an interesting observation that transcription from internal enhancers can have an attenuating effect on a host gene's transcription. Transcription can indeed be regulated by several mechanisms involving premature transcriptional termination (reviewed in Kamieniarz-Gdula and Proudfoot, Trends in Genetics, 2019). It is unclear if these and other mechanisms generalize to all TRs, and if accounting for these effects could improve the accuracy of TR prediction. These

effects might vary between different TRs and the estimation of these effects might be inaccurate with currently available data. In the revision we discuss some of the ways in which additional data might improve Lisa's accuracy.

5. Lines 100-104 & 365-366: Not sure if one-sided Wilcoxon rank sum test is appropriate here because it assumes that TRs are activators and that the computational deletion of TF cistrome is expected to reduce the model score. Two-sided Wilcoxon rank sum test should be used instead.

Lisa constructs H3K27ac ChIP-seq and DNase-seq based chromatin models which discriminate between the query gene set and the background genes. In the in silico deletion step, the assumption is that 'in silico deletion' of the regulatory TR binding sites will have the effect of weakening the discriminatory power of the model. Deleting the key regulatory elements ought to decrease the power of the model, not increase it. Therefore, the one-sided test is appropriate. We have clarified this point in the revision.

6. Line 201: "Lisa analysis of the up-regulated genes in a BCL6 knockout experiment in a DLBCL cell line ranked BCL6 first (Fig. 4b)." Explain how BCL6 is an up-regulated gene in a BCL6 knockout experiment. Shouldn't BCL6 be down-regulated in a BCL6 knockout experiment?

Both up-regulated and down-regulated genes were identified from an experiment involving the knockdown of BCL6 in the DLBCL cell line. The Lisa analysis of the up-regulated genes after BCL6 knockdown predicts BCL6 to be the most important TR. We predict BCL6 to be the regulator of the up-regulated gene set; we do not predict the BCL6 gene itself to be up-regulated. It would make sense for BCL6 to be predicted as a regulator of the up-regulated genes if BCL6 had a repressive function, which seems to be the case (see the review by Bunting and Melnick, Curr. Opinion in Immunology, 2013). Lisa's prediction is therefore consistent with the repressive function of BCL6. We have clarified this issue in the revision.

7. Lisa does well on query gene sets (Fig. 4) that are more cell type-specific and/or targets of cell type-specific TRs. The authors should test Lisa on more difficult query gene sets, which are targets of generic TRs that are ubiquitously expressed. This would widen the method's appeal to those who study basic biological processes (that are not just associated with development, organ/tissue, or diseases in particular).

In the submitted manuscript, we show that Lisa can perform well for several broadly expressed TRs, including MYC, ERG and EZH2. We have added some discussion about factors that might influence Lisa's TR prediction accuracy.

Minor Points:

1. I recommend using 'predict' instead of 'identify' in the abstract (line 21) and in the main text.

We have changed 'identify' to 'predict' in the abstract.

2. Line 59-60: "ChIP-Seq data availability is limited and many cistromes have not been produced for important TFs in many cell types (ENCODE consortium paper, 2012)". This statement was probably true back in 2012, but given the explosion of ChIP-Seq datasets, the authors may want to tone it down. Availability of antibodies with greater specificity is the limiting factor to ChIP-Seq data availability these days, which should be rightfully acknowledged.

Following the reviewer's comments, we have revised this sentence to: "ChIP-seq data availability, in terms of covered TRs and cell types, even with large contributions from projects such as ENCODE, is still sparse due to the limited availability of specific antibodies."

3. Line 67, [promoters of] actively transcribed genes

We have corrected this sentence.

4. Lines 163-169 of the section "Demonstration of Lisa on a Gata6 knockdown study": Statements containing 'small delta-RP', 'little difference', 'most significant p-values' all need to be backed up p-values (which is missing). And, results from the analyses should be provided as a supplementary table so readers can evaluate the performance of Lisa

We have reworded the revision to make quantitative statements more objective. Results of all the analyses we showed in the manuscript, including p-values, are available on the lisa web site:

http://lisa.cistrome.org/new_gallery/new_gallery.html.

5. The data used to generate Fig. 3 heatmaps should be provided as supplementary (excel/data) tables

The data used to produce heatmaps is available from the Cistrome DB. In Supplementary Table 1 we have provided references to the particular data sets used to produce this heatmap.

6. While the main text says 'BCL6 knockout,' figure 4b title says "BCL6 KD": which one is it?

We have corrected the main text in line 201 to "BCL6 knockdown".

Second round of review

Reviewer 2

The authors did an admirable job address concerns raised during the first review and have adequately addressed the major concerns raised. Additional tests of data coverage and overall accuracy of the method are welcome additions. While the new schematic in Fig. 1 is a little hard to digest at first glance, I think it does a pretty good job encapsulating the major points of the method as well as could be expected.